# BOOST THE IDENTITY-PRESERVING EMBEDDING FOR CONSISTENT VISUAL GENERATION

## ABSTRACT

Diffusion-based text-to-image (T2I) models have advanced high-fidelity content generation, but their inability to maintain subject consistency—preserving a target's identity and visual attributes across diverse scenes—hampers real-world applications. Existing solutions face critical limitations: training-based methods rely on heavy computation and large datasets; training-free approaches, while avoiding retraining, demand excessive memory or complex auxiliary modules. In this paper, we first reveal a key property overlooked in prior works that the identity-relevant signals, termed Identity-Preserving Embeddings (*IPemb*), are implicitly encoded in textual embeddings of frame prompts. To address the consistent T2I generation with the *IPemb* embedding, we propose Boost Identity-Preserving Embedding (*BIPE*), a training-free yet plug-and-play framework that explicitly extracts and enhances the *IPemb*. Its core innovations are two complementary techniques: Adaptive Singular-Value Rescaling (*adaSVR*) and Union Key (*UniK*). *adaSVR* applies singular-value decomposition to the joint embedding matrix of all frame prompts, amplifying identity-centric components (dominant matrix features) while suppressing frame-specific noise; crucially, it is integrated into every text encoder transformer layer to prevent *IPemb* dilution during non-linear feature transformations. *UniK* further reinforces consistency by concatenating cross-attention keys from all frame prompts (not just the current one), aligning the T2I backbone's image-text attention across the entire generation sequence. Experiments on the *ConsiStory+* benchmark demonstrate *BIPE* outperforms state-of-the-art methods in both qualitative and quantitative metrics. To address the gap in evaluating a broader range of scenarios with diversified prompt templates, we introduce *DiverStory*, which confirm the scalability of *BIPE*.

## 1 INTRODUCTION

In recent years, diffusion models (Song et al., 2020; Ho & Salimans, 2022) have driven remarkable advancements in the fidelity and diversity of text-conditioned generated content, spanning both static images (Ramesh et al., 2022; Saharia et al., 2022; Rombach et al., 2022) and dynamic videos (Kong et al., 2024; Blattmann et al., 2023; Wan et al., 2025). These large diffusion-based generative models demonstrate the capacity to render a broad spectrum of subjects within varied scene contexts underpinned by textual prompts. For text-to-image (T2I) diffusion models, the ability to preserve subject consistency—i.e., maintaining a target subject's core identity and visual attributes across diverse scene settings is a critical prerequisite for real-world applications. That includes animation synthesis (Hu, 2024; Guo et al., 2024), visual storytelling (Yang et al., 2024; Gong et al., 2023; Cheng et al., 2024), and text-to-video generation (Khachatryan et al., 2023; Blattmann et al., 2023), where narrative coherence relies on unbroken subject continuity. Despite these broader advancements in T2I generation, sustaining consistent subject identity and appearance across varying prompts and scene manipulations remains an unresolved challenge for existing diffusion-based frameworks.

A dominant paradigm of recent *consistent T2I generation* works relies on data- and computation-intensive training: this includes methods that train on large datasets to cluster subject identities (Avrahami et al., 2023) or learn large-scale mapping encoders to anchor subject features (Gal et al., 2023b; Ruiz et al., 2024). A critical drawback of such training-based strategies is their susceptibility to language drift (Heng & Soh, 2024; Wu et al., 2024; Huang et al., 2024), alongside their high resource overhead. To mitigate training costs, several training-free methods have achieved

promising subject consistency by exploiting shared internal activations within pre-trained T2I diffusion models (Tewel et al., 2024a; Zhou et al., 2024). While avoiding explicit retraining, they often demand extensive memory to store and manipulate intermediate activations, or rely on complex auxiliary modules to enforce consistency—limiting their scalability to real-world scenarios. A more recent contribution, 1Prompt1Story (Liu et al., 2025), addresses subject identity consistency by capitalizing on the context consistency inherent to language models. Specifically, their approach concatenates textual descriptions for all target frames into a single cohesive paragraph. During the generation of each individual frame, it dynamically adjusts the influence of descriptions from other frames—strengthening or weakening their impact based on the current frame's specific requirements. This method implicitly preserves subject identity consistency: it ensures shared access to the core subject's identity information across the entire sequence of generated frames. Our key observation is that, in text-to-image generation, cross-frame stable subject identity information is already implicitly encoded in the aggregated textual embeddings of the full frame-prompt sequence; however, existing methods do not explicitly model or reinforce this stable component. Instead of forcing all textual descriptions into a single context, we can directly search in the sequence-level embedding space for directions that are consistently present across frames and tightly related to subject identity. We refer to this previously overlooked, subject-centric stable signal as the identity-preserving embedding (*IPemb*), and treat it as an explicit intermediate representation for cross-frame consistent subject modeling.

In this work, we propose Boost Identity-Preserving Embedding (*BIPE*), which explicitly extracts and enhances *IPemb* from sequence-level text embeddings to strengthen subject consistency in narratives without modifying the underlying generative backbone. The core technique of *BIPE* is adaptive singular-value rescaling (*adaSVR*). We first apply singular value decomposition (SVD) to the joint embedding matrix formed by all frame-wise text embeddings, decomposing the sequence-level representation into a set of orthogonal "semantic directions." We then adaptively rescale the singular values to amplify directions that remain stable across frames and correspond to subject identity, while suppressing perturbation components that mainly capture frame-specific details. The reconstructed embeddings are used as new conditioning inputs for image generation, providing an explicitly enhanced and consistent identity basis for the same subject across the entire frame sequence. Notably, pre-trained text encoders rely on extensive non-linear operations that can distort or dilute identity representations during embedding extraction. To mitigate this, we integrate *adaSVR* operator into each transformer layer of the text encoder. This *per-layer* operation ensures that identity consistency is preserved throughout the entire textual embedding process, rather than only at the final output—preventing the gradual loss of *IPemb* during feature transformation. To further capitalize on the *IPemb*-augmented textual embeddings, we introduce a Union Key (*UniK*) technique, designed to enhance cross-frame consistency in the T2I model backbone. *UniK* leverages the cross-attention keys derived from the textual embeddings of all frame prompts (not just the current frame). By concatenating these frame-specific keys into an union key, we align the image-text cross-attention mechanism across the entire sequence of generated frames. This cross-frame attention alignment reinforces the propagation of identity signals across frames, thereby further enhancing the model's subject identity preservation performance.

In the experiments, we compare our method *BIPE* on an existing consistent T2I generation benchmark as *ConsiStory+* and compare it with several state-of-the-art methods (Zhou et al., 2024; Tewel et al., 2024a; Liu et al., 2025). Both qualitative and quantitative performance demonstrate the effectiveness of our method *BIPE*. And since the core mechanism of our method relies on manipulating textual embeddings, it avoids the scalability limitations that plague prior approaches. More specifically, *BIPE* exhibits two key practical advantages that address critical limitations of prior work: inherent compatibility with *long-story generation* and robust performance across *diverse prompt templates* based storytelling. To systematically validate these advantages and address the lack of dedicated benchmarks in existing literature, we introduce *DiverStory*, the new benchmarks tailored to evaluate consistency over extended or diverse template based prompt sequences with narrative continuity. Experiments on these two benchmarks further corroborate the superiority of *BIPE* in extreme storytelling cases. In summary, the main contributions of this paper are:

- To the best of our knowledge, we are the *first* to identify the existence of Identity-Preserving Embedding (*IPemb*) and explicitly extract such *IPemb* embedding in our method *BIPE* to maintain subject consistency in the consistent T2I generation. The extraction and application of *IPemb* is totally training-free and plug-and-play, thus is independent of the architecture design.

- To facilitate the extraction of the *IPemb* embedding, we further propose the *adaSVR* technique, which adaptively augment the subject identity information as it is the dominant components across the frame prompt embeddings. To further capitalize on the augmented *IPemb* textual embeddings, we introduce a Union Key (*UniK*) technique, designed to enhance cross-frame consistency in the T2I model backbone.

- Through extensive comparisons with existing consistent T2I generation approaches, we confirm the effectiveness of *BIPE* in generating images that consistently maintain identity throughout the existing *ConsiStory+* benchmark.

- To address the limitation of overly templated prompt data in existing evaluation frameworks, we propose the *DiverStory* benchmark, which uses more diverse, natural language-based prompts. This benchmark offers a more comprehensive and realistic testing framework, highlighting common challenges and shortcomings in current methods.

## 2 RELATED WORK

**T2I personalized generation.** T2I personalization (Gal et al., 2023a; Voynov et al., 2023; Zeng et al., 2024) aims to adapt a given model to generate images for a *new concept* by providing one or a few images. As a result, the adaptation model can generate various renditions of the new concept. One of the most representative methods is DreamBooth (Ruiz et al., 2023), where the pre-trained T2I model learns to bind a modified unique identifier to a specific subject given a few images. Following approaches (Kumari et al., 2023; Han et al., 2023) adhere to this pipeline and further improve the quality of the generation. A key limitation of such methods is the cumbersome fine-tuning required for each new subject. Recent advances in subject-driven image generation have shifted focus toward training identity encoders on large-scale datasets. Methods like IP-Adapter (Ye et al., 2023) and BLIP-Diffusion (Li et al., 2024a) employ an additional image encoder and novel layers to encode a subject's reference image, injecting this information into the diffusion model to enable subject-driven generation without further fine-tuning for new concepts. For DiT-based models (Peebles & Xie, 2023), Ominicontrol (Tan et al., 2024) has explored the inherent image reference capability within transformers, demonstrating that the DiT itself can function as an image encoder for subject reference. This research direction has been further advanced by subsequent works such as UNO (Wu et al., 2025), InfiniteYou (Jiang et al., 2025), and XVerse (Chen et al., 2025), with these capabilities and techniques now integrated into popular unified models (Deng et al., 2025; Ma et al., 2025).

**Consistent T2I generation.** Nowadays, there has been a research shift towards developing consistent T2I generation approaches (Wang et al., 2024a; 2025; 2024b), which can be considered a specialized form of T2I personalization. These methods mainly focus on generating human faces that possess semantically similar attributes to the input images. They mainly take advantage of PEFT techniques (Ryu, 2023; Kopiczko et al., 2024) or pre-training with large datasets (Ruiz et al., 2024; Xiao et al., 2023) to learn the image encoder to be customized in the semantic space. For example, PhotoMaker (Li et al., 2024c) enhances its ability to extract identity embeddings by fine-tuning part of the transformer layers in the image encoder and merging the class and image embeddings. However, most consistent T2I generation methods (Akdemir & Yanardag, 2024; Wang et al., 2024a) still require training the parameters of the T2I models, sacrificing compatibility with existing pre-trained community models, or fail to ensure high face fidelity. Additionally, as most of these systems (Li et al., 2024c; Ruiz et al., 2024) are designed specifically for human faces, they encounter limitations when applied to non-human subjects. Even for the state-of-the-art approaches, including StoryDiffusion (Zhou et al., 2024) and ConsiStory (Tewel et al., 2024a), they either require time-consuming iterative clustering or high memory demand in generation to achieve identity consistency. The most related prior work is 1Prompt1Story (Liu et al., 2025), which was the first to explore context consistency in language models. Its core approach concatenates all frame-specific prompts into a single sequence, leveraging this aggregated context to maintain subject identity consistency. Nonetheless, it overlooks a critical detail: *identity-relevant embeddings* are already inherently encoded within the textual embeddings of the prompt sequence itself. Additionally, the prompt concatenation mechanism faces practical limitations while being extended to long-story generation scenarios.

**Storytelling.** Story generation (Li et al., 2019; Maharana et al., 2021; Souček et al., 2025) is one of the active research directions that is highly related to character consistency. Recent researches (Tao et al., 2024; Wang et al., 2023; Zhang et al., 2025) have integrated the prominent pre-trained T2I dif-

fusion models (Rombach et al., 2022) and the majority of these approaches require intense training over story datasets. For example, Make-a-Story (Rahman et al., 2023) introduces a visual memory module designed to capture and leverage contextual information throughout the story generation.

In this paper, our proposed *BIPE* diverges significantly from previous storytelling and consistent T2I generation methods. We explore the inherent *IPemb* embedding in the text encoder instead of fine-tuning large models or designing complex modules. Importantly, it is compatible with various T2I generative models, since the properties of the text model are independent of the backbone designs.

## 3 METHODOLOGY

Consistent text-to-image (T2I) generation seeks to produce a sequence of images that depict the same subject across diverse scenes, typically using prompts that keep the subject and style descriptors similar while varying the scene descriptor (Zhou et al., 2024; Tewel et al., 2024b). Despite similar subject descriptors, base models often exhibit identity drift: different scene contexts systematically shift the embeddings of the subject token and the padding token [EoT] during text encoding—embeddings that together govern how the subject is realized in the image (Chen et al., 2023a; Li et al., 2024b). In subsection 3.1, we analyze this phenomenon and show that, despite these shifts, frame-wise text embeddings implicitly share an identity-preserving component (*IPemb*) that captures the subject's stable appearance. Building on this observation, we aim to explicitly recover and enhance this shared component at inference time, so that all frames condition on a reinforced and consistent identity signal rather than relying solely on per-frame context. To this end, we propose the Boost Identity-Preserving Embedding framework (*BIPE*), which operates directly in the text-embedding space and consists of two complementary techniques. First, adaptive singular-value rescaling (*adaSVR*, subsection 3.2) enhances the *IPemb* component within subject-related embeddings at every Transformer layer, reinforcing the shared identity representation while suppressing frame-specific fluctuations. Second, Union Key (*UniK*, subsection 3.3) concatenates the key vectors of selected tokens across prompts during cross-attention, which keeps the model's attention anchored to the same subject while preventing direct information leakage between prompts. Because *BIPE* modifies only text embeddings, the framework remains architecture-agnostic and requires neither extra data nor training. *BIPE* acts as a lightweight plug-and-play module that introduces only modest computational cost and negligible memory overhead.

### 3.1 PRELIMINARIES

**Diffusion Models.** We employ SDXL (Podell et al., 2023) as the default instantiation of *BIPE*. Its core component is a conditional U-Net $\epsilon_\theta$ (parameters $\theta$) for denoising. The training objective is:

$$L_{\text{LDM}} = \mathbb{E}_{x \sim p\text{data}, \, \epsilon \sim \mathcal{N}(0, \boldsymbol{I}), \, t \sim \mathcal{U}1,...,T} \left[; \|\epsilon - \epsilon_\theta(z_t, t, \boldsymbol{C})\|_2^2 \right], \quad (1)$$

where $z = \mathcal{E}(x)$ is the latent produced by the VAE encoder $\mathcal{E}(\cdot)$, $t$ is the timestep, and $\boldsymbol{C}$ denotes the text embeddings. SDXL uses CLIP as the text encoder $\tau_\xi$ and computes $\boldsymbol{C} = \tau_\xi(\mathcal{P}) \in \mathbb{R}^{N \times M \times D}$ from a batch of prompts $\mathcal{P} = (\mathcal{P}_1, \ldots, \mathcal{P}_N)$, where $N$, $M$, and $D$ are the batch size, number of tokens, and embedding dimension, respectively. For a given input, the denoiser $\epsilon_\theta$ fuses image-latent features with text features via cross-attention. Let $f_{z_t}$ be the feature of $z_t$ at a cross-attention block in $\epsilon_\theta$, and define queries by a projection , $\mathcal{Q} = \ell_Q(f_{z_t})$. Keys and values are obtained from the text embeddings via projections $\mathcal{K} = \ell_K(\boldsymbol{C})$ and $\mathcal{V} = \ell_V(\boldsymbol{C})$. Cross-attention is computed as:

$$\begin{aligned} \mathcal{A} &= \text{softmax}\left( \mathcal{Q}\mathcal{K}^\top / \sqrt{d} \right), \\ \mathcal{O} &= \mathcal{A}\mathcal{V}, \end{aligned} \quad (2)$$

where $d$ is the key/query dimension, $\mathcal{A}$ is the cross-attention map, and $\mathcal{O}$ is the block output.

**Problem Setup.** Consistent T2I methods compute text embeddings from a prompt set to guide subject-consistent image generation. Given $\mathcal{P} = (\mathcal{P}_1, \ldots, \mathcal{P}_N)$, we form $\mathcal{C} = [\mathcal{C}_1, \ldots, \mathcal{C}_N]$ with $\mathcal{C}_i = \tau_\xi(\mathcal{P}_i)$ for $i \in 1, \ldots, N$. Prior work often assumes that prompts follow a single template—an identical identity prefix plus a frame-specific scene description (e.g., ["A cat", "in the tree", . . . , "is sleeping"]). We refer to such prompts as *Consistent Prompts*. In contrast, we consider a broader setting in which prompts share only the same subject description while otherwise varying in sentence

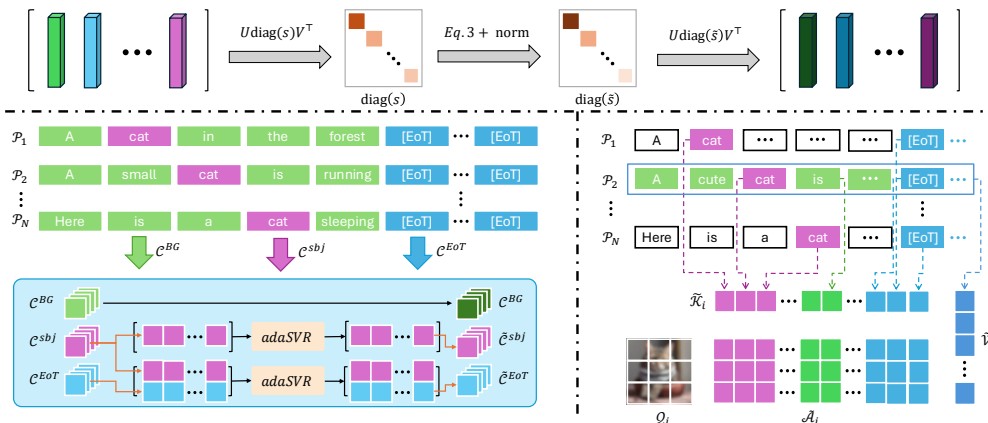

Figure 1: Overall pipeline of our method *BIPE*. (Top) the *adaSVR* operator; (Bottom-Left) *adaSVR* is applied at every self-attention layer of the text encoder, separately enhancing subject tokens and [EoT] tokens; (Bottom-Right) during cross-attention, *UniK* shares keys for specific tokens across frames while using values from the same frame. The white boxes denote the background scene tokens are not used for the current frame generation.

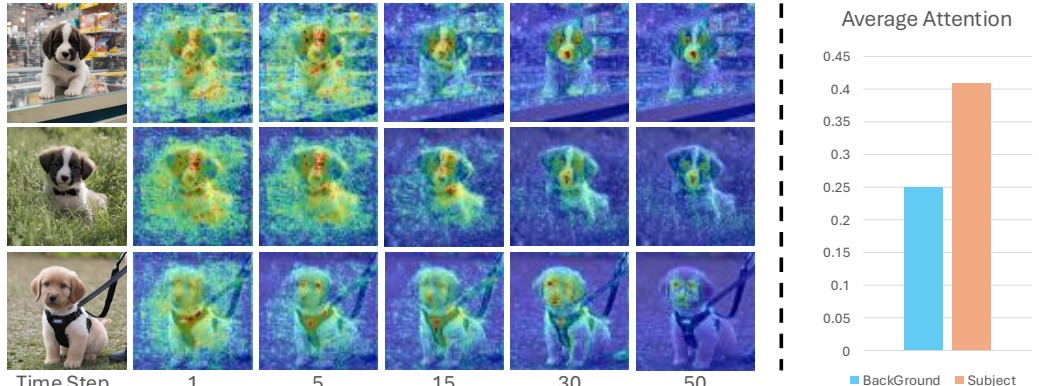

Figure 2: (Left) The leading right singular vector $v_0$ concentrates attention on the subject region across timesteps; (Right) By statistics on the *ConsiStory+*, we observe that the average masked attentions of $v_0$ still mainly focus on the subject region.

structure (e.g., ["A cat in the tree", ..., "Here is a cat sleeping"]); we term these *Diverse Prompts*. Based on the characteristics of the task, we regard the $i$-th prompt's embedding sequence as three token types, $\mathcal{C}_i = [\mathcal{C}_i^{sbj}, \mathcal{C}_i^{BG}, \mathcal{C}_i^{EoT}]$, where $\mathcal{C}_i^{sbj}$ contains subject-descriptive tokens, $\mathcal{C}_i^{BG}$ contains scene-descriptive tokens, and $\mathcal{C}_i^{EoT}$ contains padding-related tokens, including the start-of-text [SoT], end-of-text [EoT], and other padding tokens. Accordingly, we collect all subject-related tokens as $\mathcal{C}^{sbj} = [\mathcal{C}_1^{sbj}, \mathcal{C}_2^{sbj}, \dots, \mathcal{C}_N^{sbj}]$, $i \in \{1 \dots N\}$ and analogously define $\mathcal{C}^{BG}$ and $\mathcal{C}^{EoT}$.

**Identity-Preserving Embedding (*IPemb*).** In consistent T2I image generation, frames with similar subject descriptions often yield different subject identities. This largely stems from the text encoder's self-attention conditioning tokens on scene context, which induces frame-dependent shifts in the resulting text-conditioning embeddings. Meanwhile, the common subject descriptions have been encoded in the text embeddings across frames. We therefore hypothesize that per-frame text embeddings contain a shared subject-identity component that can induce consistent subject depiction. To validate this hypothesis, we extract the first [EoT] token embedding from each frame prompt—denoted $\mathcal{C}_i^{EoT}[1]$—and stack them row-wise to form $\bar{X} \in \mathbb{R}^{N \times D}$. We then apply singular value decomposition[1], $\bar{X} = \bar{U}\text{diag}(\bar{s})\bar{V}^\top$, where $\bar{s} = (s_0, \dots, s_{k-1})^\top$, $k = \text{rank}(\bar{X})$, and $\bar{V} = [v_0, \dots, v_{k-1}]$ collects the right singular vectors.

---

[1]Previous methods regard self-attention as a data-dependent linear operator on the value vectors $\mathcal{V}$ (Bhojanapalli et al., 2020; Wang et al., 2020; Geng et al., 2021; Chen et al., 2023b).

The right singular vectors associated with larger singular values (in particular, the leading vector $\boldsymbol{v}_0$ linked to $s_0$) capture shared linear patterns across frame embeddings; We use $\boldsymbol{v}_0$ as a probe token and record its cross-attention maps with the image queries $\mathcal{Q}$ during denoising in the U-Net $\epsilon_\theta$. As shown in Figure 2, the leading right singular vector $\boldsymbol{v}_0$ consistently concentrates attention on the main subject across frames, indicating that directions associated with large singular values encode a cross-frame identity-preserving embedding (*IPemb*).

### 3.2 ADAPTIVE SINGULAR-VALUE RESCALING

Inspired by our *IPemb* observation above, we need to strengthen the shared linear patterns across embeddings from different prompts. To achieve consistent T2I generation with such objective, we start by defining the Adaptive Singular-Value Rescaling (*adaSVR*) operator (see Figure 1a). The operator takes as input a matrix $\boldsymbol{X} \in \mathbb{R}^{n \times D}$ that collects a subset of text embeddings from the output of a self-attention block at some layer in the text encoder, and returns their spectrally enhanced counterpart. To start this operator, we first compute the SVD of $\boldsymbol{X} = \boldsymbol{U}\mathrm{diag}(\boldsymbol{s})\boldsymbol{V}^\top$. In this decomposition, larger singular values correspond to singular vectors that capture the shared linear patterns in $\boldsymbol{X}$, which should be emphasized. We apply an adaptive weighting to amplify such singular values:

$$\boldsymbol{w} = \exp\left(\tau \frac{\boldsymbol{s} - \mu(\boldsymbol{s})}{\sigma(\boldsymbol{s})}\right),$$

$$\hat{\boldsymbol{s}} = \boldsymbol{w} \odot \boldsymbol{s} \tag{3}$$

where $\mu(\cdot)$ and $\sigma(\cdot)$ denote the mean and standard deviation of a vector, respectively; $\tau$ is a temperature parameter that controls sensitivity to singular-value differences; and $\odot$ denotes the Hadamard (elementwise) product. This z-score–based weighting increases each singular value in proportion to its standardized magnitude while mitigating variance-induced over-amplification. Exponential weighting can over-amplify components of $\hat{\boldsymbol{s}}$, substantially increasing the reconstruction energy of $\boldsymbol{X}$. To maintain scale stability, we apply energy-matching normalization as $\tilde{\boldsymbol{s}} = \hat{\boldsymbol{s}} \cdot \frac{\|\boldsymbol{s}\|_2}{\|\hat{\boldsymbol{s}}\|_2}$. Finally, we reconstruct using the enhanced singular values, $\tilde{\boldsymbol{X}} = \boldsymbol{U}\mathrm{diag}(\tilde{\boldsymbol{s}})\boldsymbol{V}^\top$, which serves as the output of the *adaSVR* operator.

**Applying *adaSVR*** to the text encoder's final output only is insufficient due to extensive nonlinear operations within each of the encoder layers. We therefore integrate *adaSVR* into every self-attention layer and the encoder's output layer (see Figure 1b). For each such layer, we construct as:

$$\boldsymbol{X}^{EoT} = \begin{bmatrix} \mathcal{C}^{sbj} \\ \mathcal{C}^{EoT} \end{bmatrix}, \quad \boldsymbol{X}^{sbj} = \mathcal{C}^{sbj}, \tag{4}$$

and then we apply the *adaSVR* operator to obtain $\tilde{\boldsymbol{X}}^{EoT} = \mathrm{adaSVR}(\boldsymbol{X}^{EoT})$, $\tilde{\boldsymbol{X}}^{sbj} = \mathrm{adaSVR}(\boldsymbol{X}^{sbj})$. We then recover $\tilde{\mathcal{C}}^{EoT}$ from the padding rows of $\tilde{\boldsymbol{X}}^{EoT}$ and set $\tilde{\mathcal{C}}^{sbj} = \tilde{\boldsymbol{X}}^{sbj}$, yielding the enhanced sequence $\tilde{\mathcal{C}} = [\tilde{\mathcal{C}}^{sbj}, \mathcal{C}^{BG}, \tilde{\mathcal{C}}^{EoT}]$. At the output layer only, we omit the normalization step within *adaSVR* to further boost the subject signal while avoiding instability during intermediate propagation.

We apply PCA visualization to the text embeddings in Figure 3 and observe that, relative to the original embeddings, the enhanced embeddings significantly exhibit a more compact distribution in embedding space. This approach naturally extends to multi-subject generation: for each subject's description, we construct a separate subject-embedding matrix $\tilde{\boldsymbol{X}}^{sbj}$ and enhance it independently, thereby preserving subject specificity while avoiding cross-subject and cross-attribute interference.

Figure 3: Using the same prompts, we encode them with SDXL and with our *BIPE*, then visualize the resulting text embeddings via PCA. The enhanced embeddings exhibit a markedly tighter distribution in embedding space than the original SDXL embeddings.

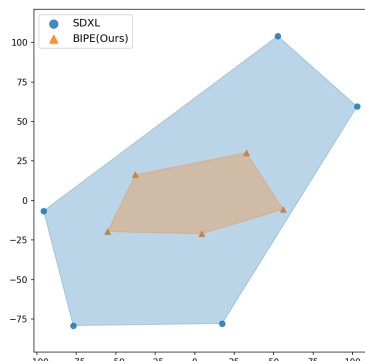

### 3.3 UNION KEY FOR CROSS-ATTENTION

To further enhance subject consistency, we introduce an attention-map-based consistency constraint, Union Key (*UniK*). The core idea is intuitive: token embeddings that are semantically equivalent

across prompts should induce the same attention distribution on the same image. For example, consider the subject embeddings $\mathcal{C}_i^{sbj}$ and $\mathcal{C}_j^{sbj}$ from prompts $i$ and $j$. If they denote the same subject, then during generation of image $j$, replacing its subject token with $\mathcal{C}_i^{sbj}$ should yield cross-attention maps (with respect to the query $\mathcal{Q}_j$) that are consistent with those obtained using $\mathcal{C}_j^{sbj}$.

Inspired by this, we introduce Union Key (As shown in Figure 1c), which computes attention using keys aggregated across prompts while applying values from the current prompt to generate the output. Specifically, for the $i$-th image, we define

$$
\begin{aligned}
\tilde{\mathcal{K}}_i &= \text{Concat}(\tilde{\mathcal{K}}_0^{sbj}, \ldots, \tilde{\mathcal{K}}_{n-1}^{sbj}, \mathcal{K}_i^{BG}, \tilde{\mathcal{K}}_0^{EoT}, \ldots, \tilde{\mathcal{K}}_{n-1}^{EoT}), \\
\tilde{\mathcal{V}}_i &= \text{Concat}(\tilde{\mathcal{V}}_i^{sbj}, \ldots, \tilde{\mathcal{V}}_i^{sbj}, \mathcal{V}_i^{BG}, \tilde{\mathcal{V}}_i^{EoT}, \ldots, \tilde{\mathcal{V}}_i^{EoT}), \\
\tilde{\mathcal{O}}_i &= \text{softmax}(\mathcal{Q}_i \tilde{\mathcal{K}}_i^{\top}/\sqrt{d})\tilde{\mathcal{V}}_i
\end{aligned}
\tag{5}
$$

where $\mathcal{Q}_i$ are the query projections for image $i$. Keys/values are obtained via linear projections from the enhanced text embeddings:

$$
\tilde{\mathcal{K}}_i^{sbj} = \ell_K(\tilde{\mathcal{C}}_i^{sbj}), \; \mathcal{K}_i^{BG} = \ell_K(\mathcal{C}_i^{BG}), \; \tilde{\mathcal{K}}_i^{EoT} = \ell_K(\tilde{\mathcal{C}}_i^{EoT}).
\tag{6}
$$

$\tilde{\mathcal{V}}_i^{sbj}$, $\mathcal{V}_i^{BG}$, and $\tilde{\mathcal{V}}_i^{EoT}$ are defined similarly. Note that the key matrix $\tilde{\mathcal{K}}_i$ is composed of subject and EoT embeddings from all frame prompts while the scene description embedding is only from the current frame $\mathcal{K}_i^{BG}$. This design is essentially equivalent to computing the attention maps of semantically aligned tokens across different prompts relative to the i-th image and averaging them, avoiding the introduction of external value vectors. By this means, we are forcing the diverse frames to share similar cross-attentions by averaging operation, which is aligned with our intuitive idea as demonstrated above. In practical applications, we assign a $1/N$ attention weight to extra K-V pairs to prevent them from dominating the image generation process. Additionally, we use only a small number of padding tokens here to keep computational costs under control. This *UniK* technique applied along with *IPemb* extracted by the *adaSVR* operator generates consistent image frames.

## 4 EXPERIMENTS

### 4.1 EXPERIMENTAL SETUP

**Comparison Methods.** We compare our *BIPE* with the following methods for consistent text-to-image generation: BLIP-Diffusion (Li et al., 2022), Textual Inversion (Gal et al., 2023a), IP-Adapter (Ye et al., 2023), PhotoMaker (Li et al., 2024c), ConsiStory (Tewel et al., 2024b), StoryDiffusion (Zhou et al., 2024), and 1Prompt1Story (Liu et al., 2025). We follow the default settings reported in their papers or open-source implementations and use 50 denoising steps for inference.

**Benchmarks.** Following prior work (Liu et al., 2025), we evaluate on the *ConsiStory+*. However, existing benchmarks typically construct data with a single template (*Consistent Prompts*), forcing all frames to share the same prefix. This introduces template bias and artificially lowers the difficulty of consistent generation. To address this, we propose the *DiverStory* benchmark: it comprises 200 carefully curated prompt sets that maintain a common subject description and a similar visual style while spanning diverse scenes; crucially, these prompts employ varied, natural-language formulations (*Diverse Prompts*) rather than a single template. Compared with existing benchmarks, *DiverStory* better reflects real user prompt distributions and reveals model consistency and robustness across a wider range of scenarios.

**Evaluation Metrics.** To assess prompt–image alignment, we compute the average CLIPScore (Hessel et al., 2021) between each generated image and its corresponding text prompt (CLIP-T) and report VQAScore (Lin et al., 2024). For identity consistency, we measure inter-image similarity using DreamSim (Fu et al., 2023) and CLIP-I (Hessel et al., 2021). Prior work shows that DreamSim correlates well with human judgments of visual similarity, while CLIP-I is the cosine similarity between CLIP image embeddings. To reduce background confounds, following (Fu et al., 2023), we remove image backgrounds with CarveKit (Selin, 2023) and replace them with random noise so that the similarity metrics focus on subject identity.

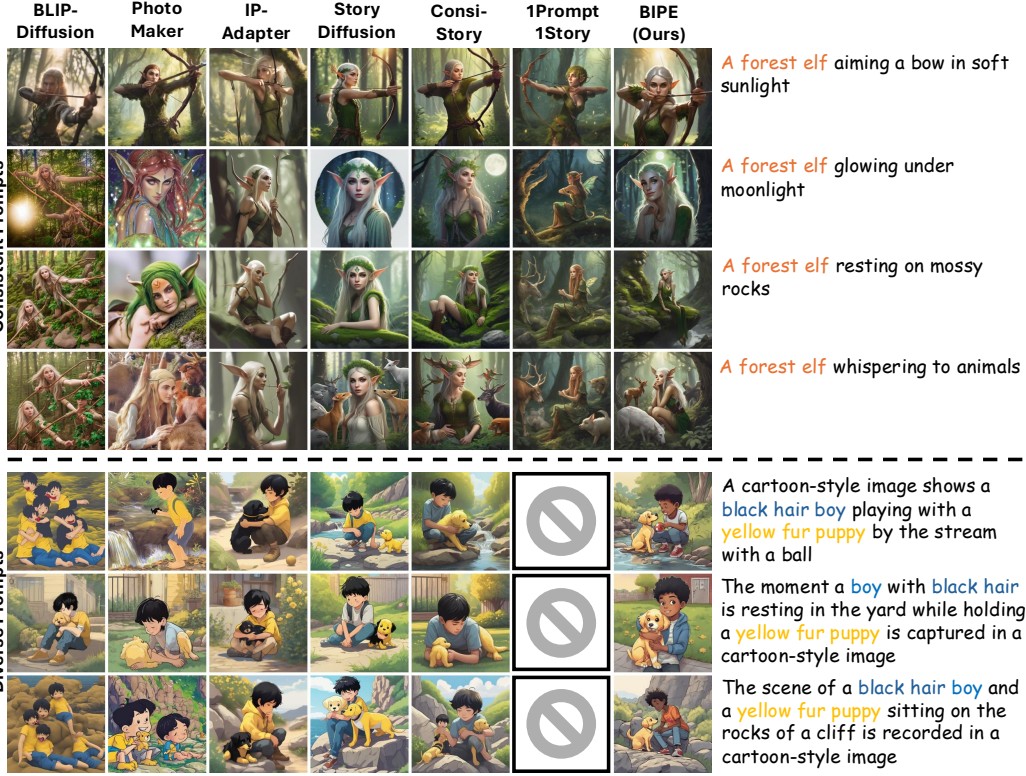

Figure 4: **Qualitative results.** We compare *BIPE* with several state-of-the-art methods. *BIPE* preserves subject-identity consistency while producing images closely aligned with the text, including background and fine-grained details. Notably, 1Prompt1Story relies on *Consistent Prompts* and does not function properly under the *Diverse Prompts* setting.

Table 1: Quantitative comparison. The best and second best results are highlighted in **bold** and underlined, respectively. Since 1Prompt1Story requires all prompts to share the same prefix, it cannot be evaluated on *DiverStory*.

| Dataset | Method | Train | CLIP-T↑ | VQA↑ | CLIP-I↑ | DreamSim↓ | Memory (GB) | Inference Time (s) |
|---|---|---|---|---|---|---|---|---|
| *ConsiStory+* | BLIP-Diffusion | ✓ | 26.84 | 0.6972 | 85.32 | 0.2624 | 8.61 | 3.89 |
| | PhotoMaker | ✓ | 30.90 | 0.8075 | 83.08 | 0.3512 | 24.70 | 19.01 |
| | IP-Adapter | ✓ | 29.76 | 0.7378 | **91.31** | **0.1654** | 19.56 | 14.16 |
| | StoryDiffusion | ✗ | 31.32 | 0.8274 | 88.58 | 0.2266 | 35.11 | 34.89 |
| | ConsiStory | ✗ | 31.27 | 0.8297 | 87.12 | 0.2438 | 46.47 | 26.61 |
| | 1Prompt1Story | ✗ | 30.11 | 0.7855 | 88.36 | 0.2153 | 18.81 | 23.51 |
| | *BIPE* (Ours) | ✗ | **31.44** | **0.8381** | 89.10 | 0.2053 | 17.16 | 20.12 |
| *DiverStory* | BLIP-Diffusion | ✓ | 26.98 | 0.6500 | 84.90 | 0.2689 | 8.61 | 3.89 |
| | PhotoMaker | ✓ | 30.93 | 0.8024 | 79.56 | 0.4208 | 24.70 | 19.01 |
| | IP-Adapter | ✓ | 29.37 | 0.7019 | **89.10** | **0.2214** | 19.56 | 14.16 |
| | StoryDiffusion | ✗ | 31.18 | 0.8220 | 84.83 | 0.3093 | 35.11 | 34.89 |
| | ConsiStory | ✗ | 31.38 | 0.8219 | 84.42 | 0.3124 | 46.47 | 26.61 |
| | *BIPE* (Ours) | ✗ | **31.85** | **0.8360** | 85.04 | 0.2918 | 17.16 | 20.12 |

## 4.2 EXPERIMENTAL RESULTS

**Qualitative Comparison.** Figure 4 presents the main qualitative comparison results. Under both the *Consistent Prompts* and our *Diverse Prompts* setups, *BIPE* delivers more balanced and stable performance across key dimensions: subject identity preservation, frame-level text–image alignment, and pose diversity. By contrast, other methods typically degrade in at least one of these aspects. More specifically, BLIP-Diffusion(Li et al., 2022) suffers from severe quality degradation, PhotoMaker (Li et al., 2024c), StoryDiffusion (Zhou et al., 2024), and ConsiStory (Tewel et al., 2024b) exhibit weak identity consistency, often introducing implausible artifacts and substantial confusion

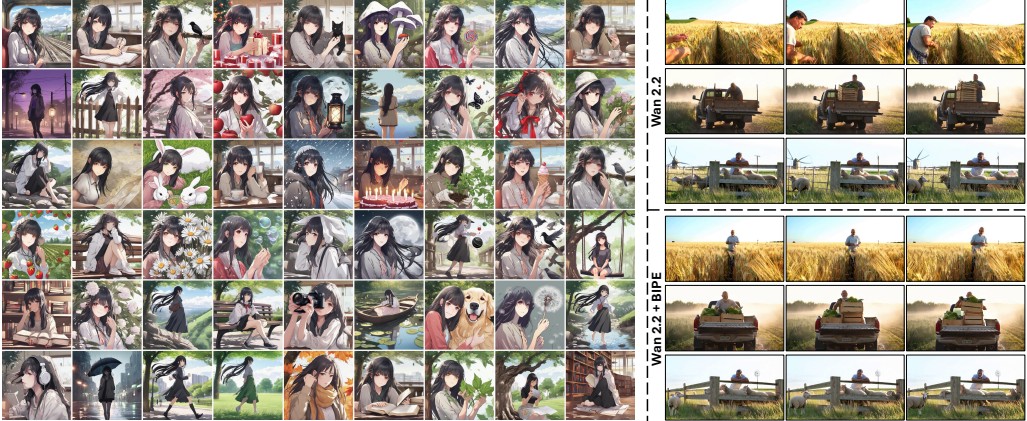

Figure 5: Qualitative ablations.

Table 2: Quantitative ablations by removing each component.

| Method | adaSVR for $\mathcal{C}^{sbj}$ | adaSVR for $\mathcal{C}^{EoT}$ | UniK | CLIP-T↑ | VQA↑ | CLIP-I↑ | DreamSim↓ |
|---|---|---|---|---|---|---|---|
| A | ✓ | ✗ | ✗ | 31.79 | 0.8460 | 86.55 | 0.2631 |
| B | ✗ | ✓ | ✗ | 31.62 | 0.8321 | 88.68 | 0.2267 |
| C | ✓ | ✓ | ✗ | 31.58 | 0.8335 | 88.80 | 0.2139 |
| D | ✗ | ✗ | ✓ | **31.84** | **0.8466** | 86.11 | 0.2686 |
| *BIPE* | ✓ | ✓ | ✓ | 31.44 | 0.8381 | **89.10** | **0.2053** |

Figure 6: **Additional applications.** (Left) *BIPE* remains stable in long-form story generation, maintaining subject identity across multiple images. (Right) Applied to state-of-the-art video generation models (Wan2.2 IT2V-5B, (Wan et al., 2025)), *BIPE* preserves consistency across multiple videos.

in multi-subject scenes. While IP-Adapter better preserves subject identity, it frequently ignores environmental and layout specifications in the text. 1Prompt1Story suffers from cross-scene contamination and mode collapse, and its requirement for *Consistent Prompts* limits applicability to more general textual inputs.

**Quantitative Comparison.** Table 1 reports quantitative comparisons with prior methods. On *ConsiStory+*, *BIPE* attains the best text–image alignment, ranks second overall in identity consistency, and is first among training-free methods. Although IP-Adapter achieves the strongest identity consistency, its text alignment degrades markedly; StoryDiffusion and ConsiStory lag on identity metrics and incur 2–3× inference overhead; and 1Prompt1Story leaves room for improvement in alignment. Compared with other approaches, *BIPE* maintains strong performance with inference speed close to the SDXL base model and does not rely on a specific prompt template, yielding broader applicability. On *DiverStory*, the ranking mirrors *ConsiStory+*, but absolute scores drop across the board (BLIP-Diffusion is an exception, albeit with noticeably degraded image quality), suggesting that current consistency methods have not yet fully extracted context-invariant identity representations and still depend, to some extent, on fixed contextual structure. Decoupling identity features from scene context remains important for future work.

**Ablation Study.** We assess component contributions via ablations, with qualitative and quantitative results shown in Figure 5 and Figure 2, respectively. When *adaSVR* is applied only to the subject description, the effect is limited due to the smaller number of subject-related tokens. However, applying *adaSVR* to both the subject description and the [EoT] token yields a significant baseline performance. Using *UniK* alone may lead to less interpretable results, as embeddings across frames lack alignment. In contrast, adding the *UniK* module on top of *adaSVR* significantly improves subject identity consistency while maintaining prompt alignment.

**Additional Applications.** Last but not the least, we extend *BIPE* to long-form stories (sequences exceeding 50 images), where it continues to deliver strong, consistent results. Moreover, since *BIPE* incurs negligible additional VRAM overhead, we further explore cross-video consistency generation—an application that has been nearly infeasible for prior methods (Figure 6).

## 5 CONCLUSION

To address the consistency T2I generation, we introduce *BIPE*, which explicitly extracts and enhances Identity-Preserving Embeddings (*IPemb*) through the adaptive singular-value rescaling (*adaSVR*) technique and reinforces cross-frame alignment via the Union Key (*UniK*) mechanism. Unlike prior approaches, *BIPE* operates in the training-free and plug-and-play manner, avoiding the scalability and resource limitations of training-based or memory-intensive strategies. Evaluations on existing benchmarks and our new *DiverStory* demonstrate the superior performance of *BIPE* in preserving subject identity across extended narratives and diverse prompt templates. By leveraging inherent identity signals in textual embeddings, this work advances T2I consistency and provides robust benchmarks for future research.

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

## A  APPENDIX: STATEMENTS

**Limitations.** This work targets training-free decoupling of subject and scene to achieve consistent image generation, offering a flexible paradigm for artistic and design applications. However, our method requires the full prompt sequence a priori and generates subject identity internally in a prompt-driven manner; it does not currently accept an explicit user-specified identity (e.g., via a reference image or external identity embedding). This constraint limits generality and extensibility. Future work will explore reference-driven consistency to enhance the identity controllability.

**Broader Impacts.** By reinforcing the shared subject representation in text encodings, *BIPE* improves subject consistency in text-to-image generation. This capability also poses risks: (i) it may be used to synthesize deceptive or misleading images, exacerbating misinformation; and (ii) when applied to public figures or copyright-/trademark-protected IP, it may raise privacy, copyright, and broader intellectual-property compliance concerns.

**Ethical Statement.** We recognize the ethical risks associated with generative models, including privacy leakage, data misuse, and the amplification or propagation of bias. All models and base weights used in this work are publicly available, and our experiments comply with their licenses and usage policies. We will release modified code and datasets to support reproducibility and external review. We also note that consistency methods can be combined with other controllable generation techniques and may be misused to synthesize misleading content (e.g., for disinformation). We therefore advocate—and support—responsible use practices.

**Reproducibility Statement.** To facilitate replication, we will release the full source code and scripts after peer review, including the implementation of *BIPE*, experimental configurations, data-processing pipelines, and instructions for obtaining and constructing the *DiverStory* dataset. All experiments were conducted on publicly available datasets. Detailed experimental settings are provided in the appendix.

**LLM Usage Statement.** We acknowledge the assistance of ChatGPT and Gemini for language polishing and improving clarity. All wording and factual content in the manuscript have been reviewed and verified by the authors.

## B  APPENDIX: BENCHMARK

Existing benchmarks for consistency generation (e.g., *ConsiStory+*) are constructed from fixed-template prompts. This design fails to reflect the diversity of natural language, introducing artificial bias and limiting applicability in real-world interactions. To address this, we introduce the diversified prompt dataset *DiverStory*. It contains approximately 200 prompt sets, each with 4–10 prompts,

categorized into seven types—animals and plants, foods, humans, techniques, fairy tales, tools, and vehicles—to ensure broad coverage. Furthermore, to evaluate consistency in more complex settings, we include multi-subject prompt sets (about 25% of the dataset), with each set containing at least two distinct subjects. This design broadens the benchmark's scope and better reflects realistic user scenarios.

Volunteers write each prompt set and annotate the subject description tokens. Other volunteers then review these annotations. We subsequently perform a manual pass over the entire dataset to filter and validate the prompts, ensuring a balanced and diverse distribution across different categories. Finally, we assign each prompt set a corresponding category label, which gives the dataset a clear structure and facilitates subsequent evaluation and extension.

## C  APPENDIX: EXPERIMENTS

**Default settings.** In *BIPE*, we set $\tau = 0.35$ in Equation 3. During generation, all frames share the same noise initialization. For *UniK*, we directly obtain the concatenated keys $\tilde{\mathcal{K}}$ and values $\tilde{\mathcal{V}}$ from the output of *adaSVR*. We construct an attention mask by assigning the columns corresponding to the concatenated segment the value $\log\left(\frac{1}{N}\right)$ and all other columns the value $0$, and add this mask to the attention logits.

In all experiments, *BIPE* and all baselines use 50 inference steps; images are generated at $1024 \times 1024$ resolution. For methods requiring a reference image, we generate the first frame with the SDXL base model and use it as the reference for subsequent frames.

**Qualitative Results.** In this section, we present additional qualitative results to further validate the effectiveness and efficiency of our proposed method *BIPE*. Figure 11 and Figure 12 provide additional qualitative comparisons against representative baseline methods. Our method, *BIPE*, consistently delivers superior visual fidelity while maintaining rapid inference. Figure 13 and Figure 14 show additional storytelling generation with our method *BIPE*.

**Seed Variety.** Because our method leaves the diffusion model's parameters unchanged, it preserves the base model's inherent ability to produce diverse appearances and backgrounds across random seeds. Concretely, with a fixed input prompt, varying only the initial noise yields multiple samples: across seeds, subject appearance and scene background differ; within a seed, frames in the sequence maintain strong subject consistency and prompt–image alignment. Figure 7 shows examples.

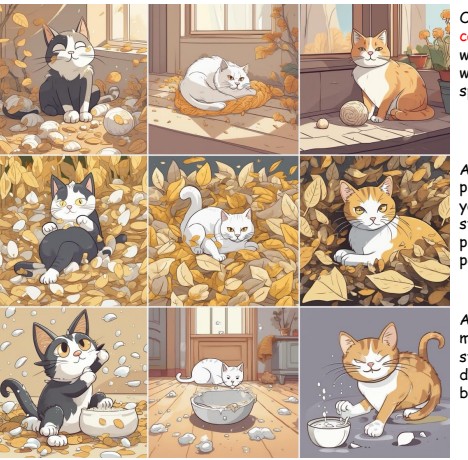

Figure 7: **Seed variation.** With fixed prompts, changing the random seed enables *BIPE* to generate images with diverse backgrounds and details while preserving subject identity consistency.

**Backbone Generalization.** We adopt the SDXL model (Podell et al., 2023) as the default backbone of *BIPE*. To demonstrate the broad compatibility of our method with different generative models, we transfer *BIPE* to Stable Diffusion 3 (Esser et al., 2024) (SD3). This model uses a joint CLIP–T5

Table 3: Quantitative results of *BIPE* implemented on SD3 on *ConsiStory+*. Since most subject-consistent image generation methods are implemented on SDXL, we directly use the native SD3 results as the baseline.

| Method | CLIP-T↑ | VQA↑ | CLIP-I↑ | DreamSim↓ |
|---|---|---|---|---|
| SD3 | 31.16 | 0.8633 | 85.66 | 0.3043 |
| SD3 + Ours | 30.63 | 0.8549 | 88.61 | 0.2114 |

text encoder and employs MM-DiT as the denoising module. As shown in Table 3, *BIPE* performs strongly on SD3. In particular, *BIPE* achieves the best text alignment and attains the second-best subject consistency among training-free methods, only behind the result of *BIPE* on SDXL. We attribute this second-best performance to the complex text-encoder architecture in SD3 and to the semantic discrepancy between CLIP and T5. We hypothesize that *BIPE* can achieve optimal performance by assigning separate $\tau$ values to CLIP and T5 and by introducing a carefully designed fusion module.

## C.1 ADDITIONAL ABLATIONS.

In this section, we conduct additional ablation studies. These experiments examine the sensitivity of *BIPE* to the temperature parameter $\tau$ in *adaSVR* and the impact of enabling or disabling attention layers. We also evaluate how the number of shared keys in *UniK* affects the performance of *BIPE*. Finally, we analyze the computational efficiency of *BIPE* under different settings.

### C.1.1 TEMPERATURE SENSITIVITY

As shown in Table 4, we report the effect of different temperatures on *BIPE* under the default setting. Overall, as the temperature increases, text alignment monotonically decreases, while subject consistency first decreases and then increases. When the temperature increases, the text embeddings become more consistent and gradually deviate from the standard distribution. With a small temperature, more consistent embeddings improve subject consistency; with a large temperature, embeddings that deviate too far from the distribution prevent the model from generating subject-stable images. We balance this trade-off and choose $\tau = 0.35$ as the default setting for the CLIP text encoder.

Table 4: Performance of *BIPE* on *ConsiStory+* under different temperatures $\tau$. † denotes the default temperature.

| *adaSVR* $\tau$ | Number of *UniK* | CLIP-T↑ | VQA↑ | CLIP-I↑ | DreamSim↓ |
|---|---|---|---|---|---|
| 0.15 | | 31.76 | 0.8523 | 87.99 | 0.2312 |
| 0.25 | | 31.70 | 0.8512 | 88.50 | 0.2156 |
| 0.35† | 10 | 31.44 | 0.8381 | 89.10 | 0.2053 |
| 0.45 | | 30.93 | 0.8133 | 88.91 | 0.2071 |
| 0.55 | | 30.04 | 0.7780 | 88.45 | 0.2154 |

### C.1.2 ATTENTION LAYER CHOICES

In this section, we investigate how different attention layers in the CLIP text encoder affect *BIPE*. Specifically, we disable the *adaSVR* enhancement operator for a subset of attention layers and observe the corresponding performance changes. In general, disabling any subset makes the text embeddings closer to the original CLIP encodings and thus reduces subject consistency. However, attention layers closer to the output have a more pronounced impact than earlier attention layers.

Table 5: In SDXL, the CLIP text encoder has 12 attention layers. We disable the *adaSVR* enhancement in the first one-third, middle one-third, and last one-third of layers, respectively, and observe the performance trends on *ConsiStory+*. The first row shows the default method without disabling any layer.

| Close Layers | *adaSVR* $\tau$ | Number of *UniK* | CLIP-T↑ | VQA↑ | CLIP-I↑ | DreamSim↓ |
|---|---|---|---|---|---|---|
| ✗ | | | 31.44 | 0.8381 | 89.10 | 0.2053 |
| 0-3 | 0.35 | 10 | 31.46 | 0.8380 | 89.10 | 0.2045 |
| 4-7 | | | 31.50 | 0.8396 | 89.02 | 0.2068 |
| 8-11 | | | 31.66 | 0.8449 | 88.38 | 0.2175 |

### C.1.3 SHARED KEY NUMBER

In this subsection, we investigate how the number and weight of shared keys in *UniK* affect performance. We vary the number of shared keys in the range [0, 77]; when the number of `[EoT]` tokens is smaller than the target, we automatically select the maximum available number of `[EoT]` tokens. Experiments on *ConsiStory+* show that *BIPE* reaches optimal performance with only a small number of shared keys, and further increasing the number has little impact on performance (as show in Table 6). This property guarantees the scalability of our method in long-story scenarios. We attribute this behavior to the low-rank nature of `[EoT]` embeddings (Li et al., 2024b), where a small number of `[EoT]` embeddings inject sufficient information. When the number of shared keys is 0, *UniK* degenerates into sharing only the subject-description embeddings.

Table 6: Performance of *BIPE* with different numbers of shared keys on *ConsiStory+*. On SDXL, only 5–10 shared `[EoT]` keys are sufficient to achieve optimal performance. † denotes the default temperature.

| *adaSVR* $\tau$ | Number of *UniK* | CLIP-T↑ | VQA↑ | CLIP-I↑ | DreamSim↓ |
|---|---|---|---|---|---|
| | 0 | 31.45 | 0.8395 | 88.93 | 0.2086 |
| | 5 | 31.43 | 0.8386 | 89.02 | 0.2053 |
| 0.35 | $10^{\dagger}$ | 31.44 | 0.8381 | 89.10 | 0.2053 |
| | 20 | 31.45 | 0.8390 | 89.05 | 0.2058 |
| | 40 | 31.44 | 0.8388 | 89.07 | 0.2058 |
| | 77 | 31.52 | 0.8387 | 89.10 | 0.2060 |

We further analyze how *UniK* affects performance when assigning different weights to shared `[EoT]` keys in *BIPE*. Specifically, we assign weight $\omega$ to keys from the current frame and weight $\frac{1-\omega}{N-1}$ to keys shared from the other frames. When $\omega = \frac{1}{N}$, this configuration is equivalent to the default setting; when $\omega = 1$, the configuration is equivalent to using zero shared `[EoT]` keys. As shown in Table 7, the resulting performance change behaves like an interpolation between the default setting and the configuration without shared `[EoT]` keys.

Table 7: Performance of *BIPE* on *ConsiStory+* with different weights $\omega$ for shared `[EoT]` keys. *BIPE* achieves strong performance by simply setting $\omega = \frac{1}{N}$.

| *adaSVR* $\tau$ | Number of *UniK* | $\omega$ | CLIP-T↑ | VQA↑ | CLIP-I↑ | DreamSim↓ |
|---|---|---|---|---|---|---|
| | | $\frac{1}{N}$ | 31.44 | 0.8381 | 89.10 | 0.2053 |
| | | 0.25 | 31.45 | 0.8387 | 89.08 | 0.2059 |
| 0.35 | 10 | 0.5 | 31.46 | 0.8379 | 89.00 | 0.2056 |
| | | 0.75 | 31.47 | 0.8384 | 88.87 | 0.2075 |
| | | 1.0 | 31.45 | 0.8398 | 88.88 | 0.2081 |

### C.1.4 COMPUTATIONAL EFFICIENCY

In this subsection, we discuss the computational cost of *BIPE*. Since *adaSVR* and *UniK* operate in the text-encoding and denoising stages respectively, we load the CLIP and U-Net modules separately, execute the two stages independently, and record the runtime and memory consumption to obtain more accurate measurements.

We first compare the computational cost of *BIPE* and SDXL under different numbers of frames $N$ (see Figure 8). In this experiment, we fix the number of shared [EoT] keys to 10. Compared with the SDXL backbone, *adaSVR* in the text-encoding stage adds about 0.2 seconds of extra encoding time per prompt, which accounts for a very small fraction of the end-to-end generation time. In the denoising stage, *UniK* increases the runtime by about 5%–20%, depending on $N$, and this overhead grows slowly as the number of frames increases. In terms of memory usage, *BIPE* does not introduce a noticeable increase at any stage and thus avoids out-of-memory failures. This efficiency arises because *adaSVR* only constructs matrices with the same size as the text embeddings for enhancement, and *UniK* only stores a small number of additional text embeddings. Overall, *BIPE* causes only minor computational overhead and does not impose a significant burden on users.

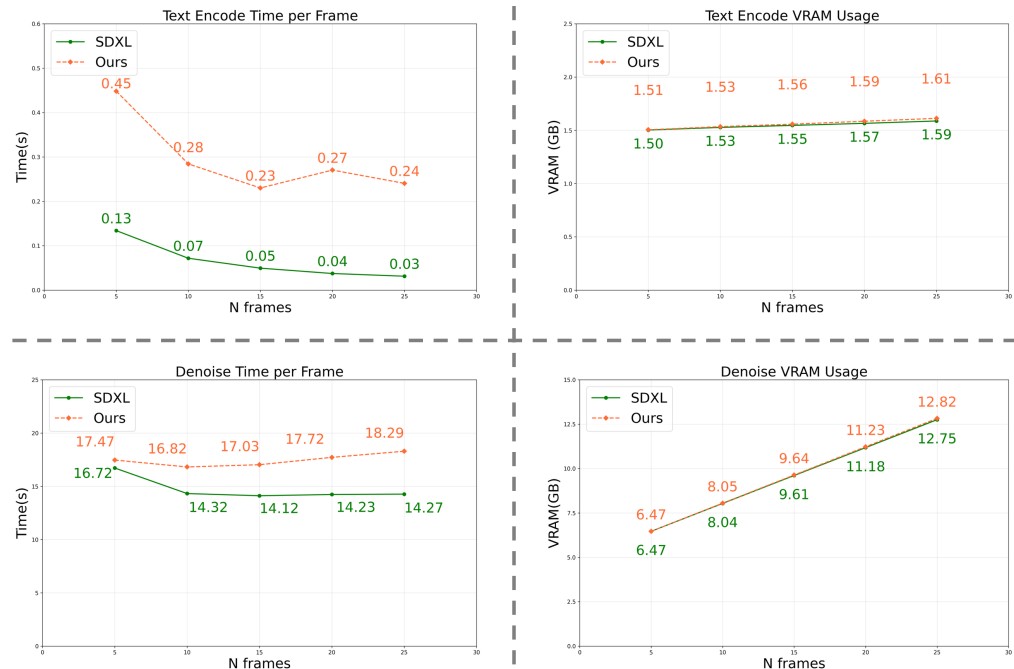

Figure 8: Computational cost of *BIPE* as the frame number $N$ varies. (Top-Left) Comparison of runtime between *BIPE* and SDXL in the text-encoding stage; as $N$ increases, *adaSVR* requires about 0.2 seconds of additional encoding time per prompt on average. (Bottom-Left) Comparison of runtime between *BIPE* and SDXL in the denoising stage; as $N$ increases, the extra time per frame is about 0.7–3 seconds. (Right) *BIPE* introduces almost no additional memory consumption.

We also examine how different numbers of shared [EoT] keys affect the computational cost of *BIPE*. As shown in Figure 9, increasing the number of shared [EoT] keys leads to a slow growth in denoising time, while the peak memory usage remains almost unchanged. Even when sharing all [EoT] keys, *BIPE* incurs only about 10% additional computation time compared with sharing only a small number of [EoT] keys.

### C.2 USER STUDY

To verify the consistency between quantitative metrics and visual quality, we conduct a user study. In the questionnaire, we compare *BIPE* with ConsiStory (Tewel et al., 2024b), StoryDiffusion (Zhou et al., 2024), and 1Prompt1Story (Liu et al., 2025). The questionnaire contains 18 prompt sets and evaluates the four methods in terms of image quality, text alignment, and subject consistency. For

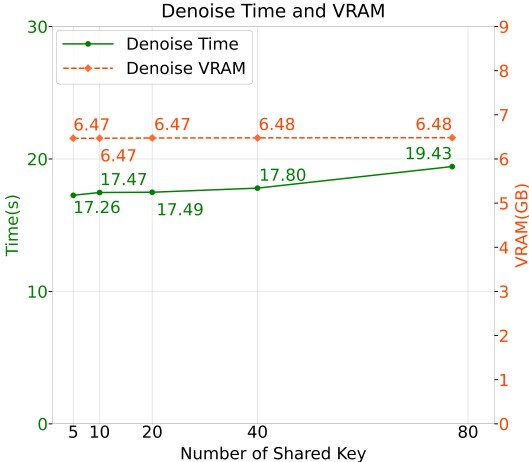

Figure 9: In the denoising stage, inference time and memory usage with different numbers of shared keys. In this experiment, we fix the number of frames at $N = 5$.

each question, participants choose the more appropriate set from two sets of generated images. We collect 67 valid responses in total (Table 8, Figure 10). On average, more than 70% of participants consider *BIPE* superior along multiple aspects, which supports the effectiveness of the proposed approach.

| Compare Methods | Image quality (win rate, %) | Text alignment (win rate, %) | Subject consistency (win rate, %) |
|---|---|---|---|
| BIPE \ ConsiStory | 64.18 \ 35.82 | 61.94 \ 38.06 | 80.60 \ 19.40 |
| BIPE \ StoryDiffusion | 73.13 \ 26.87 | 70.89 \ 29.11 | 78.35 \ 21.65 |
| BIPE \ 1Prompt1Story | 79.85 \ 20.35 | 76.11 \ 23.89 | 76.86 \ 23.14 |

Table 8: User study results. Win rate (%) of *BIPE* over each baseline in pairwise comparisons on image quality, text alignment, and subject consistency. Results are averaged over 18 prompt sets and 67 participants.

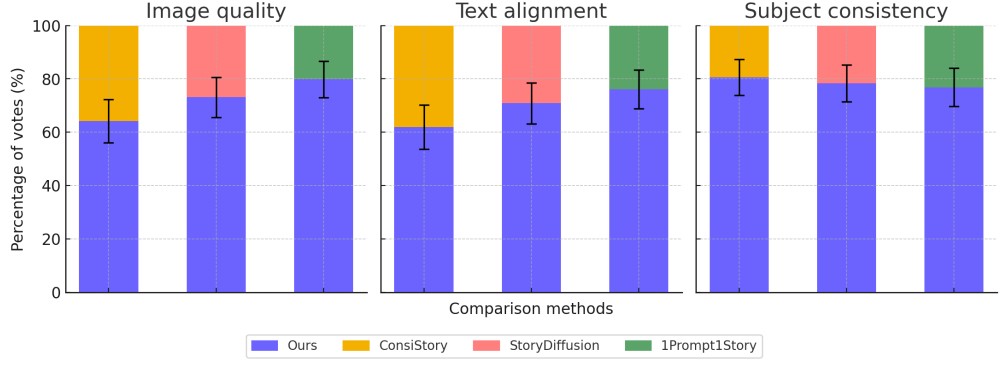

Figure 10: User-study preference scores. For each metric (image quality, text alignment, and subject consistency), the stacked bar shows the percentage of votes for *BIPE* (bottom, purple) and the corresponding baseline (top, colored). Error bars indicate the 95% confidence interval of the win rate of *BIPE*.

## C.3 ADDITIONAL APPLICATIONS.

Since our method is mainly operating on the text embeddings, we are easy to extend *BIPE* to long-form stories (sequences exceeding 50 images), where it continues to deliver strong, consistent results. Such detailed generations are demonstrated in Figure 15, Figure 16 and Figure 17.

In addition, we apply *BIPE* to multi-video consistency generation. Following the official Wan 2.2 (Wan et al., 2025) workflow, we first construct a concise set of initial prompts, then expand them using the released prompt-expansion code together with DeepSeek so that the final descriptions satisfy constraints on environment, lighting, camera, and composition. Comparative results are shown in Figure 18 and Figure 19. For readability, only the initial prompts are displayed in the figures (the expanded prompts are used for generation).

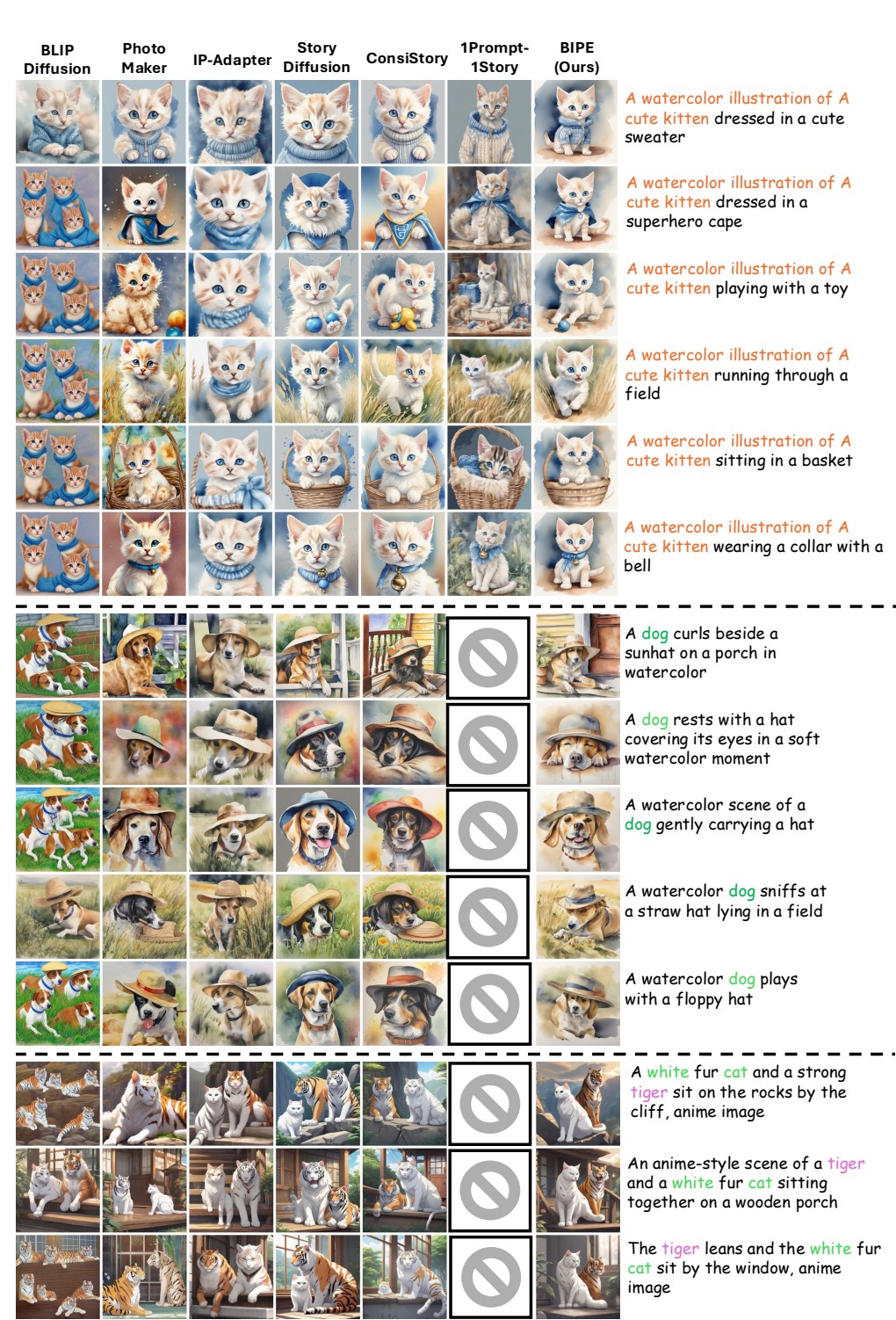

Figure 11: **Additional qualitative results.** We compare *BIPE* with several state-of-the-art methods. *BIPE* preserves subject-identity consistency while producing images closely aligned with the text, including background and fine-grained details.

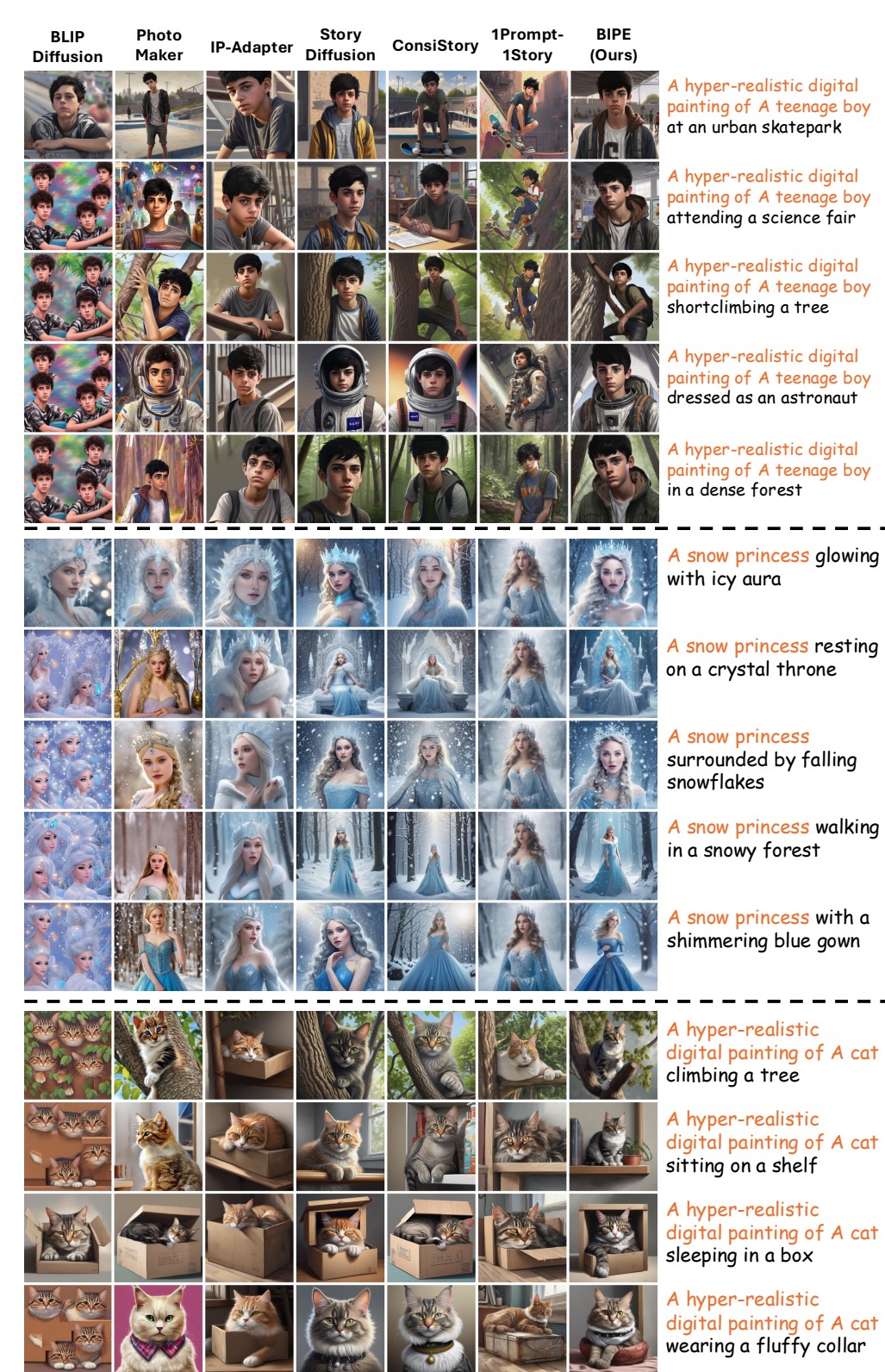

Figure 12: **Additional qualitative results.** We compare *BIPE* with several state-of-the-art methods. *BIPE* preserves subject-identity consistency while producing images closely aligned with the text, including background and fine-grained details.

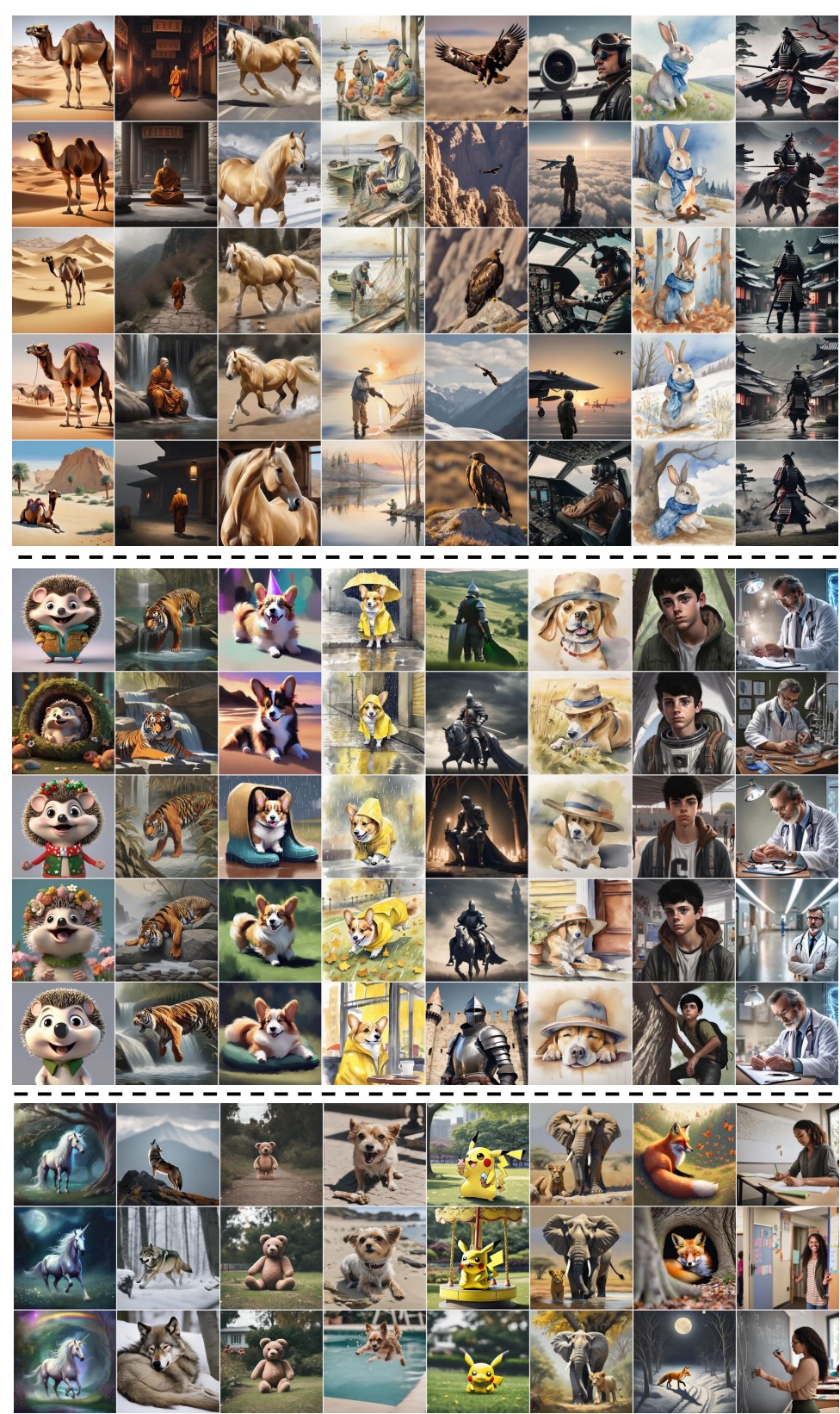

Figure 13: Additional consistent T2I generation results of *BIPE*. The vertical direction shows the same identity.

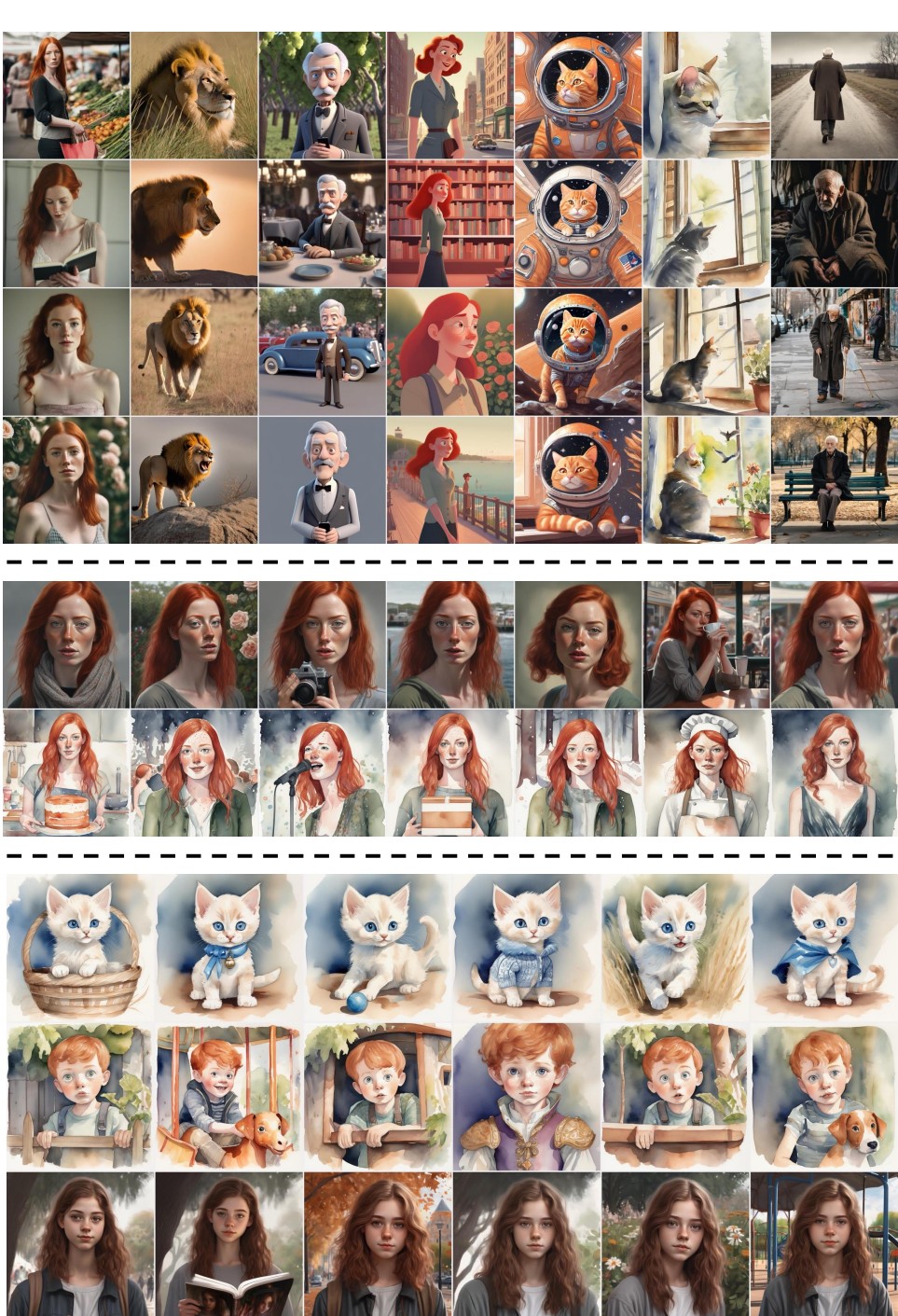

Figure 14: Additional consistent T2I generation results of *BIPE*. Note that the middle and bottom parts are showing stories horizontally.

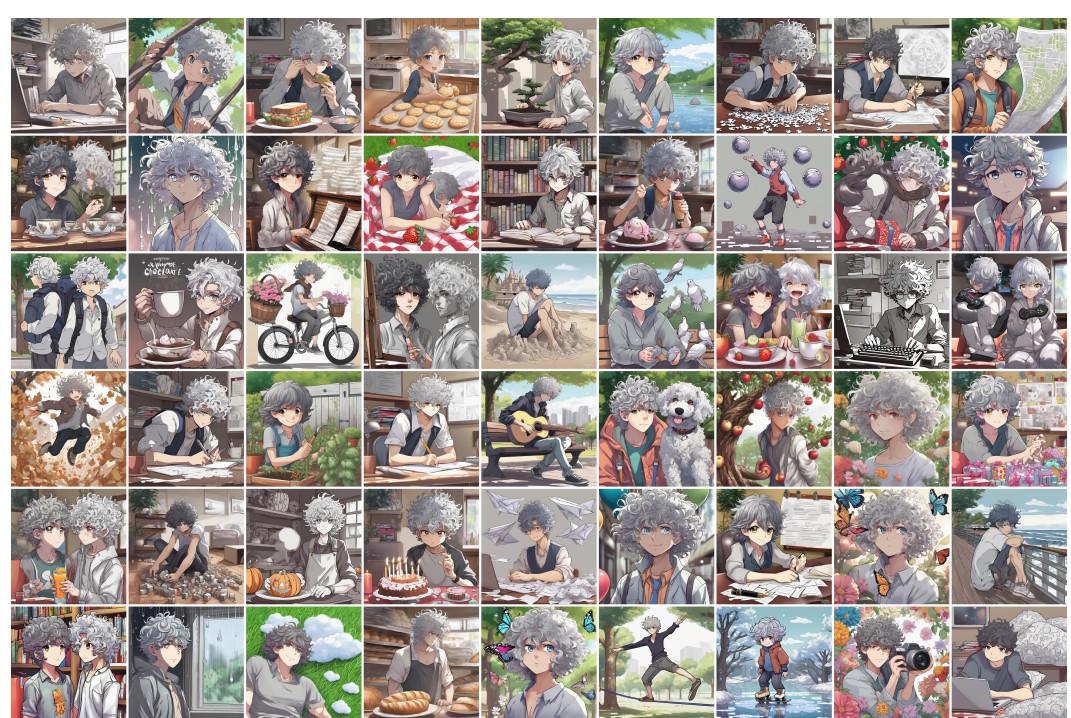

Figure 15: Long story generation results of *BIPE*.

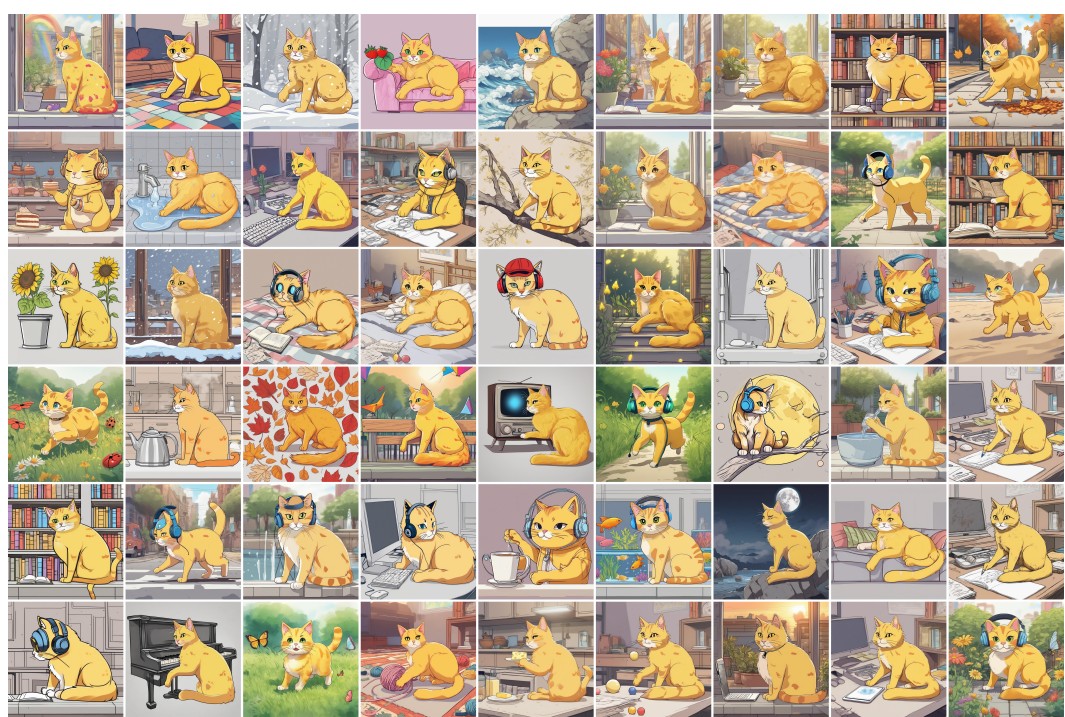

Figure 16: Long story generation results of *BIPE*.

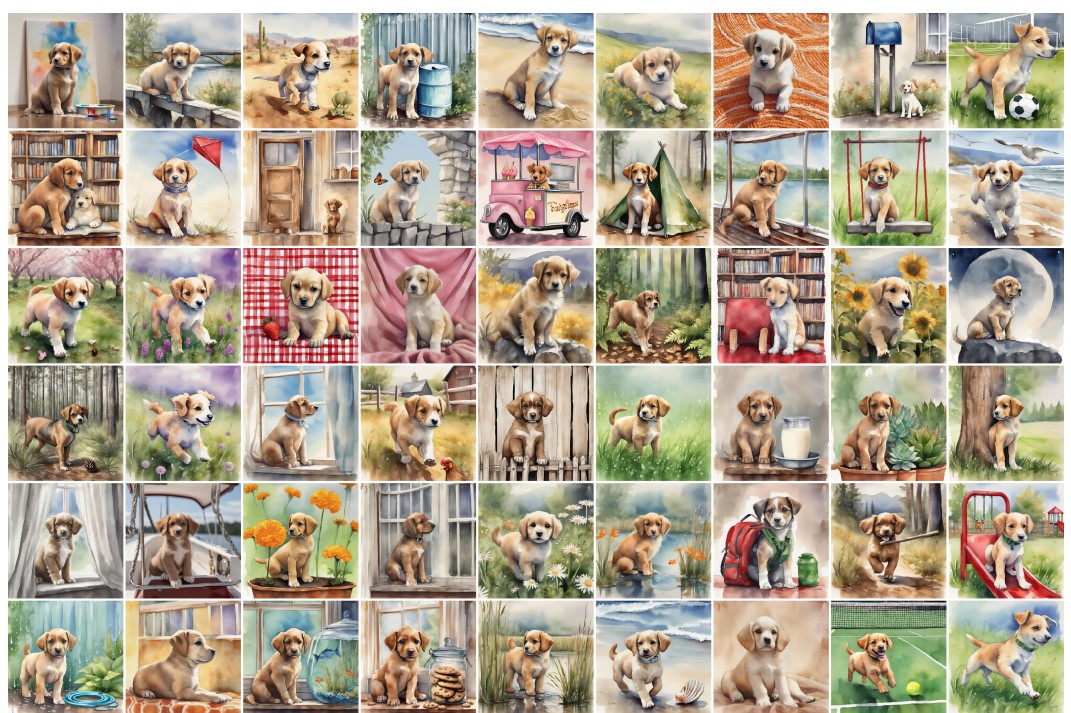

Figure 17: Long story generation results of *BIPE*.

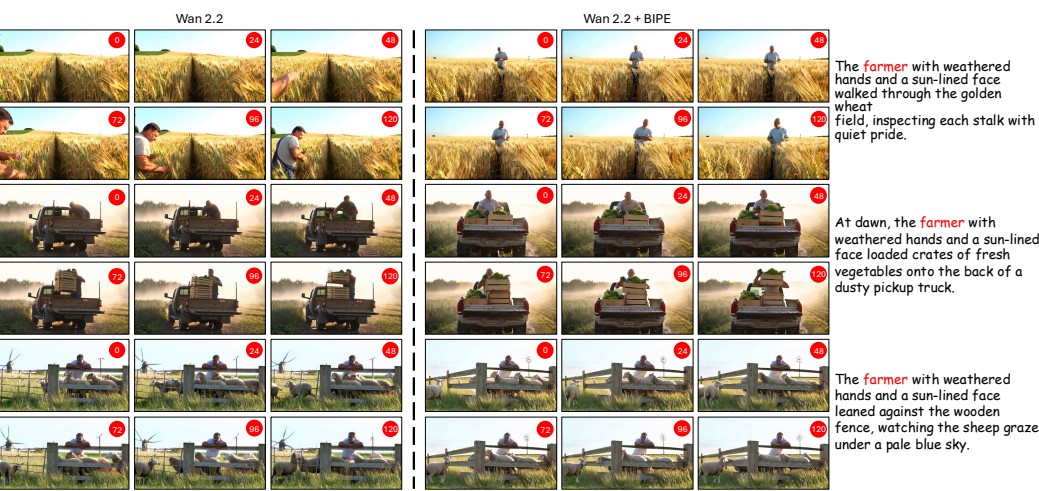

Figure 18: *BIPE* integrated into Wan2.2 enables cross-video subject-consistent generation. Frame indices are indicated by the labels.

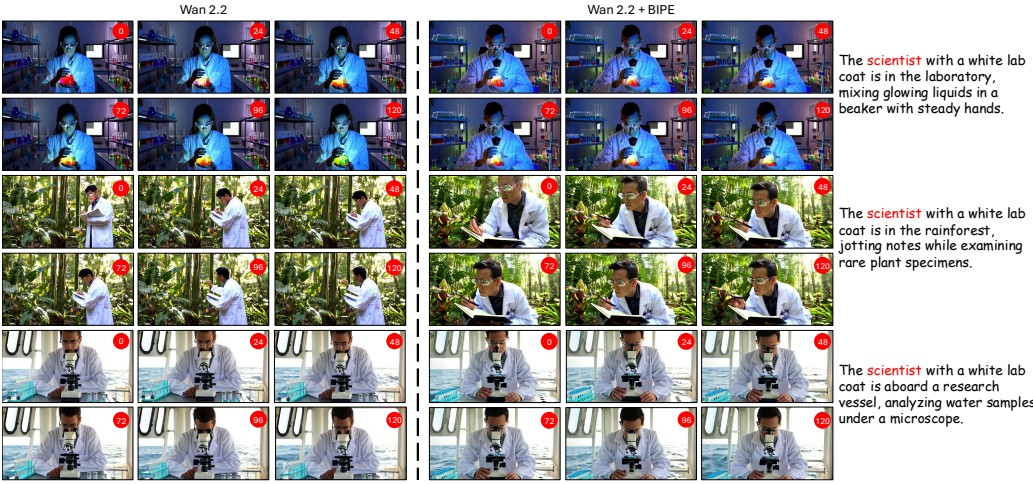

Figure 19: Another set of video generation results with *BIPE* integrated into Wan2.2, which enables cross-video subject-consistent generation. Frame indices are indicated by the labels.

