# OpenReview forum: "Boost the Identity-Preserving Embedding for Consistent Text-to-Image Generation"
_ICLR.cc/2026/Conference — Submitted to ICLR 2026_

### Official Review · Reviewer_5Bcj · 2025-10-21

**Soundness:** 3
**Presentation:** 1
**Contribution:** 2
**Rating:** 4
**Confidence:** 3

**Summary:**

The authors propose BIPE (Boost Identity-Preserving Embedding), a training-free for consistent text-to-image generation. The approach focuses on identity-preserving embeddings (IPemb) and introduces two techniques: Adaptive Singular-Value Rescaling (adaSVR) and Union Key (UniK). AdaSVR applies singular value decomposition to amplify identity-related components. UniK enhances consistency by concatenating cross-attention keys from all frame prompts. BIPE uses SDXL as the base model and is evaluated on ConsiStory+ and a newly proposed DiverStory benchmark.

**Strengths:**

1. The paper analyzes and visualizes the relationship between identity-preserving embeddings and attention mechanisms focused on the subject.
2. The authors design the DiverStory benchmark, which employs varied natural language prompt formulations rather than relying on a single fixed template as in ConsiStory+.
3. The paper provides numerous visual examples to illustrate results.

**Weaknesses:**

1. The motivation is not entirely clear. The authors claim that previous works overlook the fact that identity-relevant embedding components are already implicitly encoded within the aggregated textual embeddings of a full frame-prompt sequence. However, this limitation does not seem particularly significant, nor is it obvious that it would strongly affect results.
2. The description of the method is difficult to follow and not clearly structured. It required substantial time and effort to understand the novelty of BIPE and how it differs from 1Prompt1Story. Nevertheless, the proposed approach appears quite similar to 1Prompt1Story. For instance, the UniK component in BIPE seems analogous to Prompt Consolidation (PCon) in 1Prompt1Story, as both combine all prompts. Likewise, adaSVR in BIPE appears to resemble Singular-Value Reweighting (SVR) in 1Prompt1Story. The primary difference seems to lie in the explicit use of IPemb in BIPE. However, the distinction between explicit use in adaSVR and implicit use in SVR is not clearly explained.
3. The paper contains several typos. For example, $\bar{V}_i$ should be $\tilde{V}_i$ in Eq. (5). In Table 1, “Train-Free” should be “Train”, or “√” and “×” should be interchanged.

**Questions:**

Since BIPE is applied to video generation in the experiments, would it be more accurate to use the term “consistent visual generation” rather than “consistent text-to-image generation”?

---

> ### Author Response · Authors · 2025-11-24
> **Responses to Reviewer 5Bcj [Part 1/3]**
>
> Hi, 5Bcj,
>
> Hope you are well.
>
> We sincerely thank you for your careful reading of our manuscript and for recognizing both the visualization results and the DiverStory dataset. At the same time, your suggestions on the paper title have significantly helped us improve the manuscript. We have revised the paper based on these comments and respond to your questions one by one. All corresponding changes are highlighted in blue in the updated version of the manuscript.
>
> ## Weaknesses & Questions
> **1. The motivation is not entirely clear. The authors claim that previous works overlook the fact that identity-relevant embedding components are already implicitly encoded within the aggregated textual embeddings of a full frame-prompt sequence. However, this limitation does not seem particularly significant, nor is it obvious that it would strongly affect results.**
>
> We thank the reviewer for raising this question and would like to further clarify our motivation. Previous methods do not explicitly exploit IPemb, which forces them to rely on complex image-patch attention mechanisms (ConsiStory, StoryDiffusion). These designs introduce 2–3× overhead in memory and computation and make it difficult to scale to video and other long-horizon applications. Other methods, such as 1Prompt1Story, depend on restrictive prompt assumptions to enforce subject consistency, which leads to cross-frame information leakage and limits both performance and extensibility.
>
> Our core idea is to explicitly leverage IPemb to align text embeddings with the visual subject, and thus avoid costly architectural changes or additional assumptions. When we aggregate subject/[EoT] embeddings over all frame prompts and perform spectral decomposition, the dominant singular directions reveal a shared low-rank component that ***(a)*** remains stable across frames and ***(b)*** consistently focuses on the main subject in the image (*Fig. 2*). We refer to this component as the identity-preserving embedding (*IPemb*), which offers a new perspective on subject consistency in this domain.
>
> Our method explicitly strengthens IPemb by amplifying the IPemb-dominated singular directions at every Transformer layer, so that subject-related semantics remain stable and consistent. *Fig. 3* shows that the enhanced frame-wise text embeddings become closer. The ablation in *Tab. 2* shows that removing the IPemb-based *adaSVR* module degrades identity consistency, and that UniK alone is insufficient for subject-consistent generation. These experiments demonstrate the importance of IPemb and its decisive role in enforcing subject consistency in image generation.
>
> (To be continued.)

---

> ### Author Response · Authors · 2025-11-24
> **Responses to Reviewer 5Bcj [Part 2/3]**
>
> **2. The description of the method is difficult to follow and not clearly structured. It required substantial time and effort to understand the novelty of BIPE and how it differs from 1Prompt1Story. Nevertheless, the proposed approach appears quite similar to 1Prompt1Story. For instance, the UniK component in BIPE seems analogous to Prompt Consolidation (PCon) in 1Prompt1Story, as both combine all prompts. Likewise, adaSVR in BIPE appears to resemble Singular-Value Reweighting (SVR) in 1Prompt1Story. The primary difference seems to lie in the explicit use of IPemb in BIPE. However, the distinction between explicit use in adaSVR and implicit use in SVR is not clearly explained.**
>
> We acknowledge that BIPE and 1Prompt1Story share some technical elements (*e.g.*, both rely on singular-value-based enhancement). However, the two methods are driven by fundamentally different motivations, which in turn lead to substantial differences in input design, enhancement strategies, and scalability. We clarify these points below:
>
> ***a. Motivation.***
> 1Prompt1Story is motivated by a linguistic perspective: natural language often encodes subject consistency across context (*e.g.*, “A cat is in a tree, then it jumps onto the table.”, where “it” naturally refers to “cat”). This assumption does not rely on any specific text-encoder architecture. In contrast, BIPE is motivated by a property of Transformer-based text encoders: the same subject token experiences semantic drift across frames due to different contexts, yet attention mechanisms still preserve a shared, consistent subject component in the transformed embeddings. We empirically verify this phenomenon. This property arises from current mainstream encoder architectures rather than from an inherent property of natural language.
>
> ***b. Technical differences.***
> Because 1Prompt1Story enforces contextual consistency, it must concatenate all frame prompts into a single context, whereas BIPE does not impose this constraint.
>
> Concretely, given prompts {“A cat on a tree”, “A cat on the table”}, 1Prompt1Story uses the same merged prompt “A cat, on a tree, on the table” for both frames (*PCon*), while *BIPE* keeps the two prompts separate and uses them frame-wise. With a shared seed and a shared prompt, 1Prompt1Story effectively makes different frames identical (including the subject) and tends to produce cluttered backgrounds.
>
> This input design leads to a fundamental difference between SVR and adaSVR. 1Prompt1Story uses SVR to enhance scene information for the current frame and to suppress scene information from other frames. BIPE, instead, uses adaSVR to extract and strengthen IPemb and thereby improve subject consistency. Algorithmically, SVR keeps the subject tokens unchanged and focuses on adjusting scene tokens and [EoT] tokens, while adaSVR keeps scene tokens unchanged and aligns subject tokens and [EoT] tokens.
>
> Although *UniK* methods aggregate tokens across prompts, *BIPE* deliberately uses cross-frame information in a more restrained way. 1Prompt1Story injects all prompts directly at the input level. BIPE, through *UniK*, only shares subject-token keys and [EoT]-token keys across prompts, based on our assumption about attention-map consistency (Sec. 3.3). Moreover, BIPE never shares value vectors across frames, which reduces information contamination. Empirically, 1Prompt1Story shows lower text alignment (CLIP-T and VQA scores in *Tab. 1*), largely because it inputs all prompts without restriction.
>
> ***c. Application.***
> Since 1Prompt1Story forces all frames into a single context, it expects all prompts to share a common prefix (Consistent Prompts) and cannot naturally handle Diverse Prompts. The maximum sequence length of the text encoder further limits its use on Long Story and video scenarios.  BIPE does not suffer from these constraints. BIPE accepts fully diverse prompts and scales to long or video-like narratives, which enables broader application in realistic user scenarios.
>
> (To be continued.)

---

> ### Author Response · Authors · 2025-11-24
> **Responses to Reviewer 5Bcj [Part 3/3]**
>
> **3. The paper contains several typos. For example, $\bar{V}_i$ should be $\tilde{V}_i$ in Eq. (5). In Table 1, “Train-Free” should be “Train”, or “√” and “×” should be interchanged.**
>
> We thank the reviewer for carefully checking the manuscript. We have re-examined the corresponding parts and corrected these errors in the revised version.
>
> **4. Since BIPE is applied to video generation in the experiments, would it be more accurate to use the term “consistent visual generation” rather than “consistent text-to-image generation”?**
>
> We thank the reviewer for this suggestion and fully agree with the wording. We have revised the title accordingly to make the overall presentation of the paper more accurate and coherent.
>
> We would once again like to express our sincere gratitude to the reviewer for the time and effort devoted to our work, and for the recognition and constructive feedback. We hope that the point-by-point responses above, together with the corresponding revisions in the manuscript, have clearly addressed all of your concerns. If you have any further questions or suggestions, we would be very happy to continue the discussion and make additional improvements in the final version. We sincerely hope that, after this round of revision, the manuscript will receive your further approval.
>
> Best regards,
>
> The Authors

---

> > ### Comment · Reviewer_5Bcj · 2025-11-27
> >
> > Thanks to the authors for the detailed rebuttal, which addresses most of my concerns. This rebuttal helps me better understand the difference between BIPE and 1Prompt1Story. The ablation results in Table 2 and Figure 5 also demonstrate the effectiveness of IPemb. However, I still have some further concerns:
> >
> > 1. The authors claim that 1Prompt1Story cannot naturally handle Diverse Prompts, but a LLM could pre-convert Diverse Prompts to Consistent Prompts without much time or cost, which may reduce the significance of BIPE.
> > 2. Although BIPE differs from 1Prompt1Story in three aspects, the results without IPemb still seem acceptable, and IPemb does not appear to bring a substantial or significant improvement in visual quality.
> >
> > Therefore, I am not certain whether the novelty of this paper meets the bar for ICLR. For now, I might tend to keep my score.

---

> > > ### Author Response · Authors · 2025-12-03
> > > **Responses to Reviewer 5Bcj [Part 1/2]**
> > >
> > > Hi, 5Bcj
> > >
> > > Thank you for your further response. We are glad that most of your concerns have been resolved.
> > > Here, we would like to elaborate on the importance of Diverse Prompts and on the visual quality of BIPE.
> > >
> > > **1. The authors claim that 1Prompt1Story cannot naturally handle Diverse Prompts, but a LLM could pre-convert Diverse Prompts to Consistent Prompts without much time or cost, which may reduce the significance of BIPE.**
> > >
> > > Following your suggestion, we rewrote the DiverStory dataset using a large language model.
> > > Specifically, we used DeepSeek-V3-0324 and asked it to extract the shared style and subject descriptions, as well as the frame-specific scene descriptions, from the DiverStory inputs, and then reconstruct semantically faithful prompts from these factors.
> > > We also provided the JSON format used in ConsiStory+ and 10 example Consistent Prompts as references, and required the model to produce structured outputs.
> > > We treat the resulting prompts as a separate dataset, which we name Divers2Consis, and evaluate both BIPE and 1Prompt1Story on this dataset.
> > >
> > > |Methods|Dataset|DreamSim↓|CLIP-I↑|CLIP-T↑|VQA↑|
> > > |-|-|-|-|-|-|
> > > |1Prompt1Story|Divers2Consis|0.2381|86.64|29.41|0.7737|
> > > |BIPE|Divers2Consis|0.2190|87.94|31.26|0.8001|
> > > |BIPE|DiverStory|0.2918|85.04|31.85|0.8360|
> > >
> > > Empirically, BIPE achieves better text alignment (CLIP-T, VQA) than 1Prompt1Story while also improving subject consistency (DreamSim, CLIP-I).
> > > Compared with running BIPE directly on DiverStory, rewriting Diverse Prompts into Consistent Prompts significantly improves subject consistency but noticeably degrades text alignment.
> > > We attribute this to two factors: Consistent Prompts, which share a common prefix, make enforcing subject consistency easier than under Diverse Prompts; meanwhile, the rewriting process introduces additional semantic drift, which in turn harms text faithfulness.
> > >
> > > However, we would like to emphasize that simply using an LLM to rewrite Diverse Prompts into the Consistent Prompts format and then using them to guide generation is not a universally reliable solution.
> > > This pipeline faces substantial challenges in semantic stability and in mitigating hallucinations in model outputs.
> > > Moreover, always relying on templated prompts does not help broaden the scope of exploration and evaluation in this field.
> > >
> > > 1. Complexity of natural-language semantics and LLM hallucination.
> > >
> > > Natural language exhibits rich syntactic structures and uses different constructions to express distinct meanings precisely.
> > > Forcing these structures into a fixed format (Consistent Prompts) inevitably leads to semantic loss.
> > >
> > > In ***Fig. 4***, we present a set of Diverse Prompts as an example, whose prompts are:
> > >
> > >   1. A cartoon-style image shows a black hair boy playing with a yellow fur puppy by the stream with a ball.
> > >   2. The moment a boy with black hair is resting in the yard while holding a yellow fur puppy is captured in a cartoon-style image.
> > >   3. The scene of a black hair boy and a yellow fur puppy sitting on the rocks of a cliff is recorded in a cartoon-style image.
> > >
> > > These prompts involve two distinct subjects (the boy and the puppy) and encode different interaction patterns across frames (1: the boy plays with the puppy; 2: the boy holds the puppy).
> > > These interaction changes make it difficult to impose a single shared subject-description prefix.
> > >
> > > We attempt to rewrite the prompts using an LLM and obtain:
> > >
> > >    1. A lighthearted cartoon-style illustration of a boy with black hair and a yellow fur puppy playing with a yellow fur puppy by a stream.
> > >    2. A lighthearted cartoon-style illustration of a boy with black hair and a yellow fur puppy resting in a yard while holding a yellow fur puppy.
> > >    3. A lighthearted cartoon-style illustration of a boy with black hair and a yellow fur puppy sitting on cliffside rocks with a yellow fur puppy.
> > >
> > > To encode different interaction patterns, the LLM must reuse a shared prefix while re-describing the subjects at the end of each prompt. In practice, this often causes the generative model to produce an incorrect number of subjects (e.g., two puppies).
> > > This behavior suggests that Diverse Prompts represent a broader and more complex regime that is closer to everyday natural expressions.
> > >
> > > Moreover, when handling complex semantics, large language models often hallucinate and introduce semantic shifts during rewriting (for example, mistakenly rewriting subject attributes as scene attributes).
> > > This behavior indicates that current LLMs still cannot reliably extract the common subject and reconstruct each distinct scene when the same subject appears across multiple different scenarios.
> > >
> > > (To be continued.)

---

> > > > ### Author Response · Authors · 2025-12-03
> > > > **Responses to Reviewer 5Bcj [Part 2/2]**
> > > >
> > > > 2. Diverse Prompts provide a new perspective for the field.
> > > >
> > > > In current image generation practice, using LLMs for prompt engineering to improve visual quality has become standard.
> > > > However, directly generating from raw prompts remains crucial for assessing the core capability of a generative model.
> > > >
> > > > As shown in Tab. 1 and in the table above, existing subject-consistent generation methods still rely on context-induced scene encodings in the prompts and have not truly decoupled the subject from the scene.
> > > > This observation reveals an inherent limitation of current approaches and offers a useful perspective for future work.
> > > >
> > > > In summary, Diverse Prompts are not merely a rewritten variant of Consistent Prompts, but define a broader evaluation and application scenario, and constitute a necessary step toward real progress in subject-consistent image generation.
> > > > In line with the positive feedback from reviewers woAH and NEC8 on this perspective, we hope the reviewer will also recognize the importance and value of Diverse Prompts.
> > > >
> > > > **2. Although BIPE differs from 1Prompt1Story in three aspects, the results without IPemb still seem acceptable, and IPemb does not appear to bring a substantial or significant improvement in visual quality.**
> > > >
> > > > We address this concern by analyzing ablations and by comparing BIPE against 1Prompt1Story, focusing on visual identity and text alignment.
> > > >
> > > > |Method|DreamSim↓|CLIP-I↑|CLIP-T↑|VQA↑|
> > > > |-|-|-|-|-|
> > > > |Ablation D|0.2686|86.11|31.84|0.8466|
> > > > |BIPE|0.2053|89.10|31.44|0.8381|
> > > > |1Prompt1Story|0.2153|88.36|30.11|0.7855|
> > > >
> > > > The table above reports quantitative results on the ConsiStory+ dataset under different settings of our method and for 1Prompt1Story. Row 1 shows the result of setting D in our ablation ***Tab. 2***, where IPemb is not used for consistency enhancement.
> > > >
> > > > Compared with the variant without IPemb, full BIPE achieves a notable improvement in subject consistency (Dreamsim, CLIP-I), which demonstrates the effectiveness and importance of IPemb.
> > > >
> > > > Compared with 1Prompt1Story, BIPE further improves subject consistency while also yielding substantially better text alignment (CLIP-T, VQA). Our experiments on the Diverse2Consis dataset show the same trend.
> > > > We also note that 1Prompt1Story underperforms not only BIPE but also ConsiStory and StoryDiffusion in text alignment (CLIP-T and VQA, see Tab. 1). This degradation arises because 1Prompt1Story forces heterogeneous contexts to be concatenated into a single prompt, which often causes the generated backgrounds to merge information from multiple frames and become visually entangled.
> > > > Qualitative comparisons in Fig. 4 and Fig. 9 support this observation.
> > > > These results indicate that explicitly enhancing IPemb in BIPE provides clear advantages over constraining subject embeddings indirectly through context concatenation as in 1Prompt1Story.
> > > >
> > > > We thank the reviewer for the continued interest in our work and the constructive suggestions.
> > > > We sincerely hope that the above responses have resolved your concerns and that the manuscript will receive your further support.
> > > >
> > > > Best regards,
> > > >
> > > > The Authors

---

### Official Review · Reviewer_NEC8 · 2025-10-31

**Soundness:** 2
**Presentation:** 3
**Contribution:** 2
**Rating:** 4
**Confidence:** 4

**Summary:**

The paper tackles identity preservation in text-to-image (T2I) diffusion. It observes that a cross-frame, identity-bearing direction exists in the text-encoder embeddings. Building on this, the authors propose BIPE, a training-free, plug-and-play framework with two parts: Adaptive Singular-Value Rescaling (adaSVR) and Union Key (UniK). The paper also proposes DiverStory, a benchmark using varied natural-language prompts (not a single template), and reports gains on ConsiStory+ and DiverStory with moderate runtime/memory overhead.

**Strengths:**

- Operating purely on text embeddings makes BIPE easy to attach to SDXL-like pipelines; the paper also shows a video case (Wan 2.2).
- The IPemb observation (leading singular directions capture identity) is plausible and supported by attention-map probes.
- On ConsiStory+, BIPE achieves the best CLIP-T and VQA, with identity metrics close to the best and better efficiency than training-heavy baselines; ablations indicate complementary roles for adaSVR and UniK.
- DiverStory highlights robustness to varied natural-language prompts which is a realistic setting often under-tested.

**Weaknesses:**

- Evidence suggests BIPE helps on both template-based and diverse prompts, but several core claims and implementation details are insufficiently justified (see “Questions”).
- The empirical methodology is mostly standard, but ablations don’t fully isolate design choices (e.g., sensitivity to the weighting temperature, the role of per-layer SVD).

**Questions:**

1. Is any finetuning performed anywhere (text encoder/adapters)? If truly training-free, please correct the Table-1 flag; if not, specify what is trained and where.
2. Do UniK keys/values come from adaSVR-enhanced embeddings (as in the main text) or the original embeddings (as suggested in the appendix)? Please standardize and report the performance delta between the two setups.
3. Provide per-layer SVD dimensions and a wall-clock/VRAM profile that separates the costs of adaSVR vs. UniK, and how these scale with number of frames (N) and subject token count.
4. Include sweeps for the temperature (\tau) in Eq. (3) and the number of frames (N); additionally, report robustness to token selection (([EoT]) vs. subject tokens) and layer-wise on/off.
5. Any human studies on identity consistency under Diverse Prompts? What is the release timeline/spec for DiverStory to enable community verification?

---

> ### Author Response · Authors · 2025-11-24
> **Responses to Reviewer NEC8 [Part 1/3]**
>
> Hi, NEC8,
>
> Hope you are well.
>
> We sincerely thank you for your careful reading of our manuscript and for recognizing the motivation of our work, the performance and extensibility of our method, and the DiverStory dataset. At the same time, your suggestions on the ablation study have significantly helped us improve the manuscript. We have revised the paper based on these comments and respond to your questions one by one. All corresponding changes are highlighted in blue in the updated version of the manuscript.
>
> ## Weaknesses & Questions
> **1. Is any finetuning performed anywhere (text encoder/adapters)? If truly training-free, please correct the Table-1 flag; if not, specify what is trained and where.**
>
> Thank you for carefully reviewing the manuscript, and we apologize for these mistakes. We re-examined the paper and identified several errors in *Table 1* and related parts, which we have now corrected. We clarify that BIPE is a training-free method that relies entirely on the base model and does not train any additional components.
>
> **2. Do UniK keys/values come from adaSVR-enhanced embeddings (as in the main text) or the original embeddings (as suggested in the appendix)? Please standardize and report the performance delta between the two setups.**
>
> We apologize for the incorrect description in the appendix. We have revised the text to clarify that all text embeddings used in *UniK* are obtained from adaSVR-enhanced embeddings. In fact, during inference, our method neither computes nor stores the original SDXL text embeddings.
>
> **3. Provide per-layer SVD dimensions and a wall-clock/VRAM profile that separates the costs of adaSVR vs. UniK, and how these scale with number of frames (N) and subject token count.**
>
> In adaSVR, we first enhance the [EoT] tokens. For each column of [EoT] features, we construct an enhancement matrix of size $[len(C^{EoT}), 2N, dim]$, where $len(C^{EoT})$ denotes the number of [EoT] tokens in a single prompt, $N$ denotes the number of frames, and $dim$ denotes the intrinsic feature dimension of CLIP. We then construct an enhancement matrix for each subject of size $[len(C^{sbj}), dim]$. The SVD dimensionality follows directly from the data dimensions of these matrices; we do not impose any additional truncation or low-rank approximation.
>
> | N frame | Method       | Denoise VRAM (GB)   | Text Encode VRAM (GB) | Denoise Time per Frame (s) | Text Encode Time per Frame (s) |
> | ------: | ------------ | -------------- | ---------------- | ---------------------- | -------------------------- |
> |       5 | SDXL \ BIPE | 6.47 \ 6.47   | 1.50 \ 1.51     | 16.72 \ 17.47         | 0.13 \ 0.45               |
> |      10 | SDXL \ BIPE | 8.04 \ 8.05   | 1.53 \ 1.53     | 14.32 \ 16.82         | 0.07 \ 0.28               |
> |      15 | SDXL \ BIPE | 9.61 \ 9.64   | 1.55 \ 1.56     | 14.12 \ 17.03         | 0.05 \ 0.23               |
> |      20 | SDXL \ BIPE | 11.18 \ 11.23 | 1.57 \ 1.59     | 14.23 \ 17.72         | 0.04 \ 0.27               |
> |      25 | SDXL \ BIPE | 12.75 \ 12.82 | 1.59 \ 1.61     | 14.27 \ 18.29         | 0.03 \ 0.24               |
>
> Following the reviewer’s suggestion, we conducted computational-efficiency experiments and analysis for BIPE in ***App. C.1.4***, with results summarized in the table above. Because *adaSVR* and *UniK* operate in the text-encoding and denoising stages respectively, we separately measure the runtime and VRAM usage of these two stages. On average, adaSVR introduces about 0.2 seconds of extra encoding time per prompt, and this overhead does not change significantly with the number of frames $N$. At the same time, *adaSVR* contributes almost no additional VRAM usage, because the enhancement matrices remain small. Although this cost is noticeable relative to the bare text-encoding time, it becomes negligible compared with the overall image-generation time (about 20 seconds). In typical subject-consistent generation settings, the number of subject tokens is small (about 1–3), so we can treat it as a constant that does not materially affect computational efficiency.
>
> (To be continued.)

---

> ### Author Response · Authors · 2025-11-24
> **Responses to Reviewer NEC8 [Part 2/3]**
>
> **4. Include sweeps for the temperature ($\tau$) in Eq. (3) and the number of frames (N); additionally, report robustness to token selection (([EoT]) vs. subject tokens) and layer-wise on/off.**
> We appreciate the reviewer’s insightful question.
>
> | adaSVR $\tau$  | DreamSim↓ | CLIP-I↑ | CLIP-T↑ | VQA↑   |
> | -------------- | --------- | ------- | ------- | ------ |
> | 0.15           | 0.2312    | 87.99   | 31.76   | 0.8523 |
> | 0.25           | 0.2156    | 88.50   | 31.70   | 0.8512 |
> | 0.35 $\dagger$ | 0.2053    | 89.10   | 31.44   | 0.8381 |
> | 0.45           | 0.2071    | 88.91   | 30.93   | 0.8133 |
> | 0.55           | 0.2154    | 88.45   | 30.04   | 0.7780 |
>
> We add a temperature ablation in ***App. C.1.1***, with representative results shown above. As the temperature $\tau$ increases, text alignment (CLIP-T and VQA) decreases monotonically, while subject consistency (CLIP-I and DreamSim) follows an inverted-U trend. By balancing these two aspects, we set the default temperature to $\tau = 0.35$.
>
> | Method | adaSVR for $C^{sbj}$ | adaSVR for $C^{EoT}$ | UniK | DreamSim↓ | CLIP-I↑ | CLIP-T↑ | VQA↑   |
> | ------ | -------------------- | -------------------- | ---- | --------- | ------- | ------- | ------ |
> | A      | No                   | No                   | No   | 0.2631    | 86.55   | 31.79   | 0.8460 |
> | B      | No                   | Yes                  | No   | 0.2267    | 88.68   | 31.62   | 0.8321 |
> | C      | Yes                  | Yes                  | No   | 0.2139    | 88.80   | 31.58   | 0.8335 |
>
> The robustness to token selection ([EoT] vs. subject tokens) is analyzed in the main text (*Table 2, rows A–C*), and we reproduce part of the results above. During revision, we found that $C^{EoT}$ was mistakenly written as $C^{BG}$ in that table; we have corrected it in the manuscript.
>
> | Close Layers | DreamSim↓ | CLIP-I↑ | CLIP-T↑ | VQA↑   |
> | ------------ | --------- | ------- | ------- | ------ |
> | none         | 0.2053    | 89.10   | 31.44   | 0.8381 |
> | first 1/3    | 0.2045    | 89.10   | 31.46   | 0.8380 |
> | middle 1/3   | 0.2068    | 89.02   | 31.50   | 0.8396 |
> | last 1/3     | 0.2175    | 88.38   | 31.66   | 0.8449 |
>
> ***App. C.1.2*** reports the effect of enabling or disabling *adaSVR* on different subsets of layers. We disable *adaSVR* in the first one-third, middle one-third, and last one-third of CLIP Transformer layers, respectively. As the table above illustrates, disabling adaSVR in early Transformer layers has little impact, whereas disabling layers closer to the output of the text encoder leads to a significant performance drop. We find this behavior intriguing and plan to further investigate how different Transformer layers contribute to subject consistency in text embeddings in future work.
>
> (To be continued.)

---

> ### Author Response · Authors · 2025-11-24
> **Responses to Reviewer NEC8 [Part 3/3]**
>
> **5. Any human studies on identity consistency under Diverse Prompts? What is the release timeline/spec for DiverStory to enable community verification?**
>
> | Compare Method        | Image quality (win rate, %) | Text alignment (win rate, %) | Subject consistency (win rate, %) |
> | --------------------- | --------------------------- | ---------------------------- | --------------------------------- |
> | BIPE \ ConsiStory     | 64.18 \ 35.82               | 61.94 \ 38.06                | 80.60 \ 19.40                     |
> | BIPE \ StoryDiffusion | 73.13 \ 26.87               | 70.89 \ 29.11                | 78.35 \ 21.65                     |
> | BIPE \ 1Prompt1Story  | 79.85 \ 20.35               | 76.11 \ 23.89                | 76.86 \ 23.14                     |
>
> Following your suggestion, we conducted a user study. This study compares BIPE with three methods (ConsiStory, StoryDiffusion, and 1Prompt1Story) on multiple prompt groups under both Consistent Prompts and Diverse Prompts, and evaluates the generated images in terms of image quality, text alignment, and subject consistency. Across the 67 collected responses, more than 70% of participants, on average, judged our method to produce better visual results than the baselines along multiple dimensions. ***App. C.2*** reports the detailed setup and full results.
>
> We plan to release the DiverStory dataset together with the source code after the manuscript is officially published, so that any researcher in the community can reproduce our results, evaluate different methods, and further extend this line of work. ***App. B*** provides a detailed specification of DiverStory.
>
> We would once again like to express our sincere gratitude to the reviewer for the time and effort devoted to our work, and for the recognition and constructive feedback. We hope that the point-by-point responses above, together with the corresponding revisions in the manuscript, have clearly addressed all of your concerns. If you have any further questions or suggestions, we would be very happy to continue the discussion and make additional improvements in the final version. We sincerely hope that, after this round of revision, the manuscript will receive your further approval.
>
> Best regards,
>
> The Authors

---

### Official Review · Reviewer_woAH · 2025-11-07

**Soundness:** 3
**Presentation:** 3
**Contribution:** 3
**Rating:** 6
**Confidence:** 3

**Summary:**

The paper proposes BIPE, a training-free, plug-and-play framework that improves subject identity consistency in multi-image text-to-image generation by operating purely on text embeddings. BIPE has two components: adaptive singular-value rescaling (adaSVR), which spectrally amplifies identity-preserving directions in subject and [EoT] token embeddings across every layer of the text encoder, and Union Key (UniK), which concatenates cross-attention keys from all prompts while using per-frame values to align attention without leaking full values across frames. Experiments on ConsiStory+ and a new Diverse Prompts benchmark, DiverStory, show strong text alignment and competitive identity consistency with low memory and runtime overhead, and the authors also illustrate integration into Wan 2.2 for cross-video consistency.

**Strengths:**

Originality is solid: rather than new networks or retraining, the work identifies and boosts an intrinsic identity-preserving component in text embeddings and enforces consistency via key-sharing in cross-attention, which is simple and broadly applicable. Quality is supported by clear math for adaSVR with energy-matched normalization, principled token selection for subject and padding tokens, and a practical 1/N weighting of extra key-value pairs to control dominance and cost. Clarity is generally high, with an end-to-end pipeline and ablations that isolate adaSVR vs UniK contributions. Significance is promising since BIPE is architecture-agnostic, requires no additional data or training, and achieves strong alignment and competitive identity metrics with near-base latency, while DiverStory broadens evaluation beyond templated prompts.

**Weaknesses:**

The paper claims BIPE is training-free, yet Table 1 marks BIPE as not training-free on both ConsiStory+ and DiverStory, which conflicts with the text and should be corrected or explained. The evaluation emphasizes SDXL as the default and shows case studies with Wan 2.2, but broader quantitative tests on additional backbones would better support the architecture-agnostic claim. Identity consistency is mostly measured by CLIP-I and DreamSim with background removal; a small human study or per-attribute identity analysis would strengthen conclusions on visual identity. Finally, while the method uses only a subset of tokens in UniK to cap compute, sensitivity to the number and type of shared tokens, and scaling with the number of frames N, is not systematically profiled.

**Questions:**

a) Please reconcile the training-free claim with Table 1, which currently lists BIPE as not training-free. If this is a typesetting error, clarify and update; if not, explain what part of BIPE requires training.

b) How does BIPE scale in runtime and memory with the number of frames and with the count of shared keys in UniK? A plot of latency and VRAM vs N and vs number of shared subject/[EoT] tokens would help practitioners.

c) Beyond SDXL and the Wan 2.2 illustration, can you report quantitative results on at least one non-CLIP text encoder or a DiT-based T2I backbone to substantiate architecture-agnostic claims.

d) Could you add sensitivity studies for adaSVR’s temperature and the decision to include [EoT] alongside subject tokens, plus an ablation on the 1/N weighting strategy.

e) DiverStory is valuable; can you provide statistics on prompt diversity and subject types, along with plans and licensing for release, so the community can reproduce and extend your results.

f) The limitations note that BIPE does not accept external identity references; can you outline how BIPE would integrate with reference-image encoders or identity embeddings while retaining the training-free property.

---

> ### Author Response · Authors · 2025-11-24
> **Responses to Reviewer woAH [Part 1/3]**
>
> Hi, woAH,
>
> Hope you are well.
>
> We sincerely thank you for your careful reading of our manuscript and for recognizing the originality and significance of our work as well as the DiverStory dataset. At the same time, your suggestions on architecture-agnosticism, ablation studies, and the user study have substantially helped us improve the manuscript. We have revised the paper based on these comments and respond to your questions one by one. All corresponding changes are highlighted in blue in the updated version of the manuscript.
> ## Weaknesses & Questions
> **(a) Please reconcile the training-free claim with Table 1, which currently lists BIPE as not training-free. If this is a typesetting error, clarify and update; if not, explain what part of BIPE requires training.**
>
> Thank you for carefully reviewing the manuscript, and we apologize for these mistakes. We re-examined the paper and identified several errors in *Table 1* and related parts, which we have now corrected. We clarify that *BIPE* is a training-free method that relies entirely on the base model and does not train any additional components.
>
> **(b) How does BIPE scale in runtime and memory with the number of frames and with the count of shared keys in UniK? A plot of latency and VRAM vs N and vs number of shared subject/[EoT] tokens would help practitioners.**
>
> | N frame | Method       | Denoise VRAM (GB)   | Text Encode VRAM (GB) | Denoise Time per Frame (s) | Text Encode Time per Frame (s) |
> | ------: | ------------ | -------------- | ---------------- | ---------------------- | -------------------------- |
> |       5 | SDXL \ BIPE | 6.47 \ 6.47   | 1.50 \ 1.51     | 16.72 \ 17.47         | 0.13 \ 0.45               |
> |      10 | SDXL \ BIPE | 8.04 \ 8.05   | 1.53 \ 1.53     | 14.32 \ 16.82         | 0.07 \ 0.28               |
> |      15 | SDXL \ BIPE | 9.61 \ 9.64   | 1.55 \ 1.56     | 14.12 \ 17.03         | 0.05 \ 0.23               |
> |      20 | SDXL \ BIPE | 11.18 \ 11.23 | 1.57 \ 1.59     | 14.23 \ 17.72         | 0.04 \ 0.27               |
> |      25 | SDXL \ BIPE | 12.75 \ 12.82 | 1.59 \ 1.61     | 14.27 \ 18.29         | 0.03 \ 0.24               |
>
> We added a series of ablation and parameter-efficiency experiments that report the runtime and memory usage of BIPE under different numbers of shared keys and different frame counts $M$. The numerical results are summarized in tables, and ***App. C.1.4*** provides the corresponding line plots. During image generation, BIPE incurs slightly higher latency than the base model (about 10%) while using a similar amount of VRAM. We therefore expect this additional cost not to become a bottleneck in practice.
>
> | Shared EoT Keys | Denoise VRAM (GB) | Denoise Time (s) |
> | :-------------- | ----------------- | ---------------- |
> | 5               | 6.47              | 17.26            |
> | 10              | 6.47              | 17.47            |
> | 20              | 6.47              | 17.49            |
> | 40              | 6.48              | 17.80            |
> | 77              | 6.48              | 19.43            |
>
> For the computational cost under different numbers of shared keys, we first clarify that, in typical consistency image generation settings, the number of subject tokens is small (1–3) and thus has negligible impact on efficiency. In our experiments, we fix the number of frames to $N=5$. As the number of shared [EoT] keys increases, the runtime grows slowly, whereas peak memory usage remains almost unchanged. Compared with the setting without [EoT] sharing, the most expensive configuration requires at most about 15% extra time. As shown by the empirical results in ***App. C.1.3***, BIPE already achieves optimal performance with only 5–10 shared [EoT] keys, so the practical computational overhead remains modest.
>
> **(c) Beyond SDXL and the Wan 2.2 illustration, can you report quantitative results on at least one non-CLIP text encoder or a DiT-based T2I backbone to substantiate architecture-agnostic claims.**
>
> | Method     | DreamSim↓ | CLIP-I↑ | CLIP-T↑ | VQA↑   |
> | ---------- | --------- | ------- | ------- | ------ |
> | SD3        | 0.3043    | 85.66   | 31.16   | 0.8633 |
> | SD3 + BIPE | 0.2114    | 88.61   | 30.63   | 0.8549 |
>
> We apply BIPE to the recent Stable Diffusion 3 model, a strong image-generation model that incorporates several novel architectural components. In particular, SD3 uses a hybrid CLIP+T5 text encoder and an MM-DiT architecture as the denoising backbone. As shown in the table, although training-free methods inevitably incur a slight drop in text alignment, BIPE achieves significantly better subject consistency (CLIP-I and DreamSim). These results support our claim that the proposed method is architecture-agnostic. We have added these experiments to ***App. C***.
>
> (To be continued.)

---

> ### Author Response · Authors · 2025-11-24
> **Responses to Reviewer woAH [Part 2/3]**
>
> **(d)Could you add sensitivity studies for adaSVR’s temperature and the decision to include [EoT] alongside subject tokens, plus an ablation on the 1/N weighting strategy.**
>
> | adaSVR $\tau$  | DreamSim↓ | CLIP-I↑ | CLIP-T↑ | VQA↑   |
> | -------------- | --------- | ------- | ------- | ------ |
> | 0.15           | 0.2312    | 87.99   | 31.76   | 0.8523 |
> | 0.25           | 0.2156    | 88.50   | 31.70   | 0.8512 |
> | 0.35 $^\dagger$ | 0.2053    | 89.10   | 31.44   | 0.8381 |
> | 0.45           | 0.2071    | 88.91   | 30.93   | 0.8133 |
> | 0.55           | 0.2154    | 88.45   | 30.04   | 0.7780 |
>
> We add a temperature ablation in ***App. C.1.1***, with results shown above. As the temperature increases, text alignment decreases monotonically, while subject consistency follows an inverted-U trend. By balancing these two aspects, we set the default temperature in *adaSVR* to 0.35.
>
> | Method | adaSVR for $C^{sbj}$ | adaSVR for $C^{EoT}$ | UniK | DreamSim↓ | CLIP-I↑ | CLIP-T↑ | VQA↑   |
> | ------ | -------------------- | -------------------- | ---- | --------- | ------- | ------- | ------ |
> | A      | No                   | No                   | No   | 0.2631    | 86.55   | 31.79   | 0.8460 |
> | C      | Yes                  | Yes                  | No   | 0.2139    | 88.80   | 31.58   | 0.8335 |
>
> The decision to enhance [EoT] tokens in *adaSVR* is studied in the main paper (*Table 2, settings A and C*), with the key numbers reproduced above. During revision we noticed that $C^{EoT}$ was mistakenly written as $C^{BG}$ in the original table and we have corrected this typo. We emphasize that enhancing [EoT] tokens is necessary. Prior work [1] has shown that subject identity is encoded not only in subject description tokens but also in [EoT] tokens.
>
> | $\omega$      | DreamSim↓ | CLIP-I↑ | CLIP-T↑ | VQA↑   |
> | ------------- | --------- | ------- | ------- | ------ |
> | $\frac{1}{N}$ | 0.2053    | 89.10   | 31.44   | 0.8381 |
> | 0.25          | 0.2059    | 89.08   | 31.45   | 0.8387 |
> | 0.5           | 0.2056    | 89.00   | 31.46   | 0.8379 |
> | 0.75          | 0.2075    | 88.87   | 31.47   | 0.8384 |
> | 1.0           | 0.2081    | 88.88   | 31.45   | 0.8398 |
>
> ***App. C.1.3*** further analyzes the effect of the number and weights of shared [EoT] keys. Our default $\frac{1}{N}$ weighting strategy is motivated by its equivalence to averaging the attention maps across frames (see Sec. 3.3). In the ablation above, we assign a fixed weight $\omega \in (0,1]$ to keys from the current frame and a weight $\frac{1-\omega}{N-1}$ to keys from the other frames, so that the total weight remains 1. The results show that BIPE’s performance under different $\omega$ values smoothly interpolates between the settings with and without shared [EoT] keys.
>
> **(e) DiverStory is valuable; can you provide statistics on prompt diversity and subject types, along with plans and licensing for release, so the community can reproduce and extend your results.**
>
> We thank the reviewer for the positive feedback on the DiverStory dataset. DiverStory contains approximately 200 prompt sets, each consisting of 4–10 prompts. These prompts are divided into seven categories—animals and plants, foods, humans, techniques, fairy tales, tools, and vehicles—to ensure broad coverage of different image-generation scenarios. We also include a subset of multi-subject prompt sets. We plan to release the *DiverStory* dataset together with the source code after the manuscript is officially published, so that any researcher in the community can test and evaluate different methods or further extend them on this dataset. More details about DiverStory are provided in ***App. B***.
>
> (To be continued.)

---

> ### Author Response · Authors · 2025-11-24
> **Responses to Reviewer woAH [Part 3/3]**
>
> **(f) The limitations note that BIPE does not accept external identity references; can you outline how BIPE would integrate with reference-image encoders or identity embeddings while retaining the training-free property.**
>
> We thank the reviewer for the suggestion on extending *BIPE*. Although BIPE cannot directly consume reference images, BIPE can combine with existing methods to support reference-image-based subject-consistent generation while remaining training-free. We outline a feasible inversion-based pipeline below.
>
> **Algorithm: Reference based Consistency Image Generation Pipeline**
> - **Inputs:** reference image $I_0$ with description $P_0$; target prompts $\{P_1, \dots, P_N\}$.
> - **Outputs:** generated target images $\{I_1, \dots, I_N\}$.
> 1. $\{\tilde{C}_0, \tilde{C}_1, \dots, \tilde{C}_N\} \leftarrow \text{adaSVR}(\{P_0, P_1, \dots, P_N\})$
> 2. $z_0 \leftarrow \text{DDIMInversion}(I_0, \tilde{C}_0)$
> 3. **for** $i = 1, \dots, N$ **do**
>     $\quad I_i \leftarrow \text{Denoise}(z_0, \tilde{C}_i)$
> 4. **return** $\{I_1, \dots, I_N\}$
>
> This pipeline still faces several potential challenges that may harm the visual quality. For instance, the enhanced embedding $\tilde{C}_0$ may no longer accurately describe the reference image $I_0$, and errors introduced by the inversion step may misalign the subject identity. We plan to conduct experiments in future work to explore the potential of this integration.
>
> **(g) A small human study or per-attribute identity analysis would strengthen conclusions on visual identity.**
>
> | Compare Methods       | Image quality (win rate, %) | Text alignment (win rate, %) | Subject consistency (win rate, %) |
> | --------------------- | --------------------------- | ---------------------------- | --------------------------------- |
> | BIPE \ ConsiStory     | 64.18 \ 35.82               | 61.94 \ 38.06                | 80.60 \ 19.40                     |
> | BIPE \ StoryDiffusion | 73.13 \ 26.87               | 70.89 \ 29.11                | 78.35 \ 21.65                     |
> | BIPE \ 1Prompt1Story  | 79.85 \ 20.35               | 76.11 \ 23.89                | 76.86 \ 23.14                     |
>
> Following your suggestion, we conducted a user study. This study compares *BIPE* with three other methods (ConsiStory, StoryDiffusion, and 1Prompt1Story) on multiple prompt groups under both Consistent Prompts and Diverse Prompts, and evaluates the generated images in terms of image quality, text alignment, and subject consistency. On average, across the 67 collected responses, more than 70% of participants judged our method to produce better visual results than the baselines along multiple dimensions. Please refer to ***App. C.2*** for the detailed setup and results.
>
> We would like to once again express our sincere gratitude for your time, effort, and constructive feedback, as well as for your positive assessment of our work. We hope that the point-by-point responses above and the corresponding revisions in the manuscript have addressed all of your concerns. If you have any further questions or suggestions, we would be very happy to continue the discussion and refine the paper accordingly in the final version. We genuinely hope that, after this round of revision, the manuscript will receive your further approval.
>
> Best regards,
>
> The Authors

---

### Author Response · Authors · 2025-12-03
**Summary of Reviews and Author Responses**

Dear PCs, SACs, ACs, and Reviewers,

To assist the newly assigned AC and reduce their workload, we provide a summary of key points from the reviews and our responses during the rebuttal period.

We thank the three reviewers (**R1 = woAH, R2 = NEC8, R3 = 5Bcj**) for their positive feedback.
The reviewers agreed that the paper has a clear motivation (R1, R2, R3),
that the method is training-free, architecture-agnostic, and easy to extend (R1, R2),
and that the DiverStory dataset has strong potential as a benchmark (R1, R2, R3).

We sincerely thank all the reviewers for their thoughtful and constructive comments. The revised part in the resubmission version is highlighted in blue for your reference.
We use W to abbreviate Weaknesses, and Q to represent Questions.

**1. Training-Free** (R1(Q1),R2(Q1Q2),R3(Q3))

We have corrected the inaccuracies in the original manuscript and clarified that our method is fully training-free.
In addition, UniK is always computed on top of the text embeddings enhanced by adaSVR (R2Q2).

**2. Computational Efficiency** (R1(Q2), R2(Q3))

In the revision, we added an efficiency comparison between SDXL and SDXL+BIPE under different numbers of frames $N$ and different shared-key settings (***App. C.1.4***).
The results show that, across different $N$, BIPE on average requires about 10% additional inference time compared with SDXL, while peak VRAM usage remains almost unchanged.
Using too many shared keys in UniK introduces a small extra time overhead, but 5–10 shared keys are already sufficient to reach near-saturated performance.

**3. Architecture-agnostic Evaluation** (R1(Q3))

We implemented BIPE on SD3 and evaluated it on the ConsiStory+ benchmark (***App. C***).
The results show that our method brings significant improvements in subject consistency.
Together with the qualitative results on Wan 2.2 (***Fig. 6***), this provides empirical support for our architecture-agnostic claim.

**4. Ablation Studies** (R1(Q4), R2(Q4))

We conducted several ablation studies to demonstrate the effectiveness of our components and the robustness of our hyperparameter choices:
- IPemb-based adaSVR is necessary, and UniK further improves performance (***Tab. 2***).
- As $\tau$ increases, text-alignment metrics consistently degrade, while consistency metrics follow an inverted-U trend and achieve the best trade-off around $\tau = 0.35$ (***App .C.1.1***).
- Turning adaSVR off layer-wise degrades performance, with layers closer to the output having a larger impact (***App .C.1.2***).
- The UniK weighting strategy is robust within a reasonable range, and only a small number of shared keys is sufficient to obtain near-optimal performance, which preserves computational efficiency (***App .C.1.3***).

**5. Human Study** (R1(W1), R2(Q5))

Our user study compares BIPE with three methods (ConsiStory, StoryDiffusion, and 1Prompt1Story) on multiple prompt groups under both Consistent Prompts and Diverse Prompts settings.
The study evaluates the generated images in terms of image quality, text alignment, and subject consistency.
Across the 67 collected responses, more than 70% of participants, on average, preferred our method over the baselines along multiple dimensions (***App. C.2***).

**6. Compare 1Prompt1Story** (R3(Q1Q2Q5Q6))

We clarified the differences between BIPE and 1Prompt1Story in the Official Comments.
In terms of motivation, 1Prompt1Story builds on contextual consistency in natural language, whereas BIPE is driven by the IPemb observation.

In terms of methodology, 1Prompt1Story enforces subject consistency by concatenating prompts, which often produces cluttered backgrounds and reduces extensibility.
In contrast, BIPE explicitly enhances aligned text embeddings via IPemb and uses cross-frame tokens in a restrained manner, thereby achieving more consistent subjects while significantly improving text alignment and preserving good extensibility.

Together with the component ablations and user-study results, these additions clarify that BIPE is not a simple variant of 1Prompt1Story, but a distinct technical route built upon the new IPemb-based observation.

**Other Questions**
- We added a description of the DiverStory dataset and our release plan (R1(Q5), R2(Q5));
- We briefly discussed a possible pipeline that combines BIPE with reference images / identity encoders while remaining training-free (R1(Q6));
- We revised the title so that it more accurately covers both image and video consistency (R3(Q4));
- We corrected several notation and typesetting errors (R1(Q1), R2(Q1Q2), R3(Q3)).

We once again thank the ACs and the three reviewers for their careful evaluation and valuable feedback.
We hope that the above responses, together with the corresponding revisions in the manuscript, have clearly addressed all concerns, further strengthened the contribution and empirical support of our work.

Best regards,

The Authors

---

### Meta-Review · Area_Chair_yjDg · 2025-12-21

**Summary:**

The reviewers’ concerns primarily centered on clarity, empirical validation, and novelty, rather than fundamental correctness of the approach. Reviewers requested clearer articulation of the training-free claim, the interaction between adaSVR and UniK, and more precise descriptions to avoid ambiguity. Additional concerns focused on computational efficiency and scalability, as well as stronger evidence supporting the architecture-agnostic claim beyond SDXL. Reviewers also emphasized the need for more comprehensive ablation studies, clearer hyperparameter analysis, and stronger evaluation, including human studies and justification of the DiverStory benchmark.

Beyond these points, I also share a concern regarding the level of methodological novelty. While the paper introduces two components—one operating on the text embedding space via spectral (PCA/SVD-style) manipulation, and another modifying cross-attention behavior—both directions have been widely explored in the customization and consistent generation literature, including prior training-free and attention-based control methods. Although the authors argue that their IPemb observation provides a unifying perspective, the extent to which this constitutes a fundamentally new mechanism, rather than a principled recombination of existing ideas, may be viewed as incremental by some reviewers.

**Reviewer Concerns:**

The rebuttal satisfactorily addressed several major reviewer concerns.
First, the authors clarified ambiguities around the training-free claim, correcting misleading phrasing and clearly explaining that the method introduces no additional training or learned parameters, and that UniK operates on embeddings enhanced by adaSVR.
Second, concerns regarding computational efficiency and scalability were addressed through new quantitative analyses showing limited inference overhead and negligible VRAM increase across different frame counts and shared-key settings.
Third, the authors provided additional evidence supporting the architecture-agnostic claim, including experiments on SD3 and qualitative results on Wan 2.2, demonstrating that the method generalizes beyond SDXL.
Fourth, the rebuttal added extensive ablation studies and hyperparameter analyses, clarifying the necessity of IPemb-based adaSVR, the complementary role of UniK, layer-wise effects, and robustness to parameter choices.
Fifth, the authors strengthened the evaluation by adding a human study and clarifying the motivation, construction, and release plan of the DiverStory benchmark.
Finally, the rebuttal clearly articulated the conceptual and methodological differences from 1Prompt1Story, arguing that BIPE follows a distinct IPemb-driven design rather than prompt concatenation.

Concerns that remain partially outstanding:

Some concerns remain partially unresolved. In particular, while the novelty of IPemb is better motivated and empirically supported, a degree of conceptual overlap with prior training-free consistency methods may still be perceived by some reviewers. In addition, although DiverStory is well motivated, its scale and external adoption remain to be validated in the longer term. These issues, however, are more about positioning and future impact rather than correctness or completeness of the current work.

**Reviewer Scores:**

Reviewer R1 (woAH):
R1’s concerns were largely about clarification, efficiency analysis, and ablations. These points were comprehensively addressed in the rebuttal with additional experiments, clearer explanations, and a human study. If R1 had participated fully in the discussion, their score would likely have increased by one level.

Reviewer R2 (NEC8):
R2 was generally positive about the contribution and raised mostly technical clarification questions regarding architecture-agnostic claims, ablations, and evaluation under diverse prompts. The rebuttal directly addressed these concerns with new experiments and clearer positioning. R2’s score would likely have possibly increased slightly, though a large change is unlikely.

Reviewer R3 (5Bcj):
R3 expressed more skepticism about novelty, particularly in relation to 1Prompt1Story, and raised concerns about whether the contribution is sufficiently distinct. While the rebuttal clarified the conceptual differences and added supporting evidence, some subjective concerns about novelty may persist. As a result, R3’s score would likely increase modestly or remain unchanged, with higher confidence but possibly still cautious.

---

### Decision · Program_Chairs · 2026-01-26

Reject